# GHOST: Hallucination-Inducing Image Generation for Multimodal LLMs

**Aryan Yazdan Parast**[1][*]   **Parsa Hosseini**[2][*]   **Hesam Asadollahzadeh**[1]   **Arshia Soltani Moakhar**[2]
**Basim Azam**[1]   **Soheil Feizi**[2]   **Naveed Akhtar**[1]
[1]The University of Melbourne   [2]University of Maryland

## Abstract

Object hallucination in Multimodal Large Language Models (MLLMs) is a persistent failure mode that causes the model to perceive objects absent in the image. This weakness of MLLMs is currently studied using static benchmarks with fixed visual scenarios, which preempts the possibility of uncovering model-specific or unanticipated hallucination vulnerabilities. We introduce **GHOST** (**G**enerating **H**allucinations via **O**ptimizing **S**tealth **T**okens), a method designed to **stress-test** MLLMs by actively generating images that induce hallucination. GHOST is fully automatic and requires no human supervision or prior knowledge. It operates by optimizing in the image embedding space to mislead the model while keeping the target object absent, and then guiding a diffusion model conditioned on the embedding to generate natural-looking images. The resulting images remain visually natural and close to the original input, yet introduce subtle misleading cues that cause the model to hallucinate. We evaluate our method across a range of models, including reasoning models like GLM-4.1V-Thinking, and achieve a hallucination success rate exceeding 28%, compared to around 1% in prior data-driven discovery methods. We confirm that the generated images are both high-quality and object-free through quantitative metrics and human evaluation. Also, GHOST uncovers **transferable** vulnerabilities: images optimized for Qwen2.5-VL induce hallucinations in GPT-4o at a 66.5% rate. Finally, we show that fine-tuning on our images mitigates hallucination, positioning GHOST as both a diagnostic and corrective tool for building more reliable multimodal systems.[1]

## 1 Introduction

Multimodal large language models (MLLMs) (Liu et al., 2024; 2023a; Bai et al., 2025; Team et al., 2025) have demonstrated impressive performance across a wide range of vision and language tasks, including image captioning, visual question answering, and multimodal reasoning. However, they suffer from a potentially critical failure mode: **object hallucination**, where the model incorrectly predicts the presence of an object that is not present in the image. Such errors raise concerns about the reliability of MLLMs in safety-sensitive applications and underscore the need for systematic methods to stress-test their visual robustness.

Existing evaluations of object hallucination in MLLMs typically rely on static benchmarks, fixed visual scenarios, and curated image sets (Li et al., 2023; Hu et al., 2023; Leng et al., 2024; Lovenia et al., 2023). While useful, these approaches constrain analysis to a fixed set of visual scenarios and fail to expose model-specific vulnerabilities. Moreover, they lack a systematic way to assess which types of images trigger hallucinations, making it difficult to determine whether these errors are isolated or indicative of deeper, structural failure modes. Identifying such patterns is crucial not only for understanding model behavior but also for mitigating these failures.

To address these limitations, we introduce GHOST (Generating Hallucinations via Optimizing Stealth Tokens), a method designed to stress-test MLLMs by synthesizing hallucination-inducing images. Given an input image and a target object, GHOST generates a visually similar image that

---

[*]Equal Contribution. Correspondence to: `phoseini@umd.edu`

[1]We release code and models at `https://github.com/sudoparsa/GHOST`.

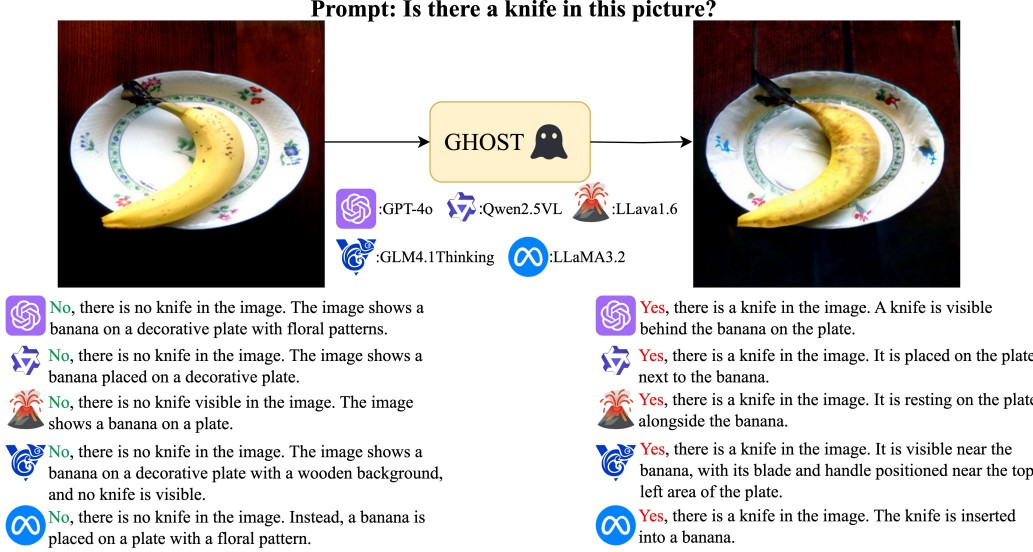

Figure 1: (**Left**) All models correctly answer "No" when asked if there is a knife in the image. (**Right**) GHOST introduces subtle cues, and all models now hallucinate the presence of a knife.

preserves the absence of the object but causes the model to incorrectly detect it. The edits introduced by GHOST are subtle and semantic, contextual cues that are perceptible to humans but do not introduce the object itself. This is accomplished by optimizing the image's CLIP embedding to elicit hallucination while regularizing against object insertion, then using it to guide the diffusion model to generate a natural-looking image.

Consider the example in Figure 1. The original image (left) shows a banana on a plate with no knife present, and multiple MLLMs correctly answer "No" when asked whether a knife is in the picture. However, after GHOST introduces subtle changes, such as modifying the banana's stem to resemble the edge of a knife, the models flip their answer to "Yes," hallucinating a knife. These misleading cues are not convincing to human observers, yet they are sufficient to push the models across the decision boundary.

One key innovation in GHOST is its design that decouples the optimization process from the image generator. Prior work on systematic failure image generation either omits feedback from the target model, limiting their ability to capture model-specific blind spots (Zhang et al., 2024; Wu et al., 2024), or uses resource-intensive and slow pipelines (Augustin et al., 2025). In contrast, GHOST introduces a mapper that aligns the visual spaces of the target model and the diffusion model, enabling efficient optimization while preserving compatibility across different image generators and vision-language models.

Together, these components make GHOST scalable and effective: for instance, on Qwen2.5-VL (Bai et al., 2025), GHOST generates 2,816 hallucination-inducing images out of 9,423 initial inputs, a 29% success rate–compared to 0.1% in DASH (Augustin et al., 2025). GHOST also generalizes to reasoning-based MLLMs like GLM-4.1V-Thinking (Team et al., 2025), where it shifts the model's reasoning trajectory to justify the presence of a nonexistent object (Figure 7). The resulting images are natural and high quality, as demonstrated by the FID results in Table 2. Figure 2 shows qualitative examples, where each column compares an original image (top) with its GHOST-edited counterpart (bottom). The subtle semantic changes introduced by GHOST cause the model to hallucinate its presence. Human evaluation confirms that GHOST images remain object-free from a human perspective, with 89% of responses agreeing the target object is absent.

GHOST also reveals transferable vulnerabilities: images optimized for one model often cause hallucinations in others. For example, images generated for Qwen2.5-VL induce hallucinations in GPT-4o at a 66.5% rate (Table 3). GHOST thus exposes not just model-specific blind spots, but systemic vulnerabilities across MLLMs.

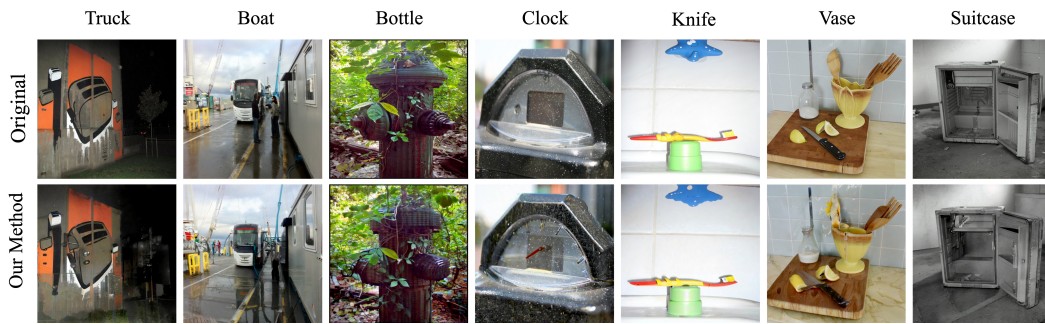

Figure 2: (**Top**) Input images, the MLLM does not hallucinate the target object. (**Bottom**) GHOST images, the MLLM hallucinates the object, despite its absence being clear to a human observer.

Finally, we demonstrate that GHOST can also aid in mitigation: fine-tuning on GHOST synthetic images improves model robustness on downstream hallucination benchmarks (Section 6.2), highlighting GHOST's potential as both a diagnostic and corrective tool.

## 2  RELATED WORKS

**Systematic Failures in MLLMs.** Recent works have introduced frameworks to automatically identify failures in MLLMs (Tong et al., 2024a;b; Hosseini et al., 2025). Tong et al. (2024b) highlight MLLMs' struggles with basic visual understanding, attributing them to limitations in CLIP-based vision encoders. In contrast, Tong et al. (2024a) focuses on failures stemming from the language modality. Hosseini et al. (2025) show that MLLMs often over-rely on spurious visual cues, leading to hallucinations when such cues are present but the object itself is absent. A large body of work has specifically examined object hallucination (Li et al., 2023; Leng et al., 2024; Lovenia et al., 2023; Hu et al., 2023; Wang et al., 2023), typically proposing benchmarks and evaluation protocols to quantify the issue. However, these approaches rely on curated image sets, constraining analysis to a fixed set of visual scenarios. In contrast, our method actively generates failure images, uncovering novel and unanticipated visuals that are unlikely to appear in static datasets.

**Discovering Hallucinations through Image Generation.** Some prior studies (Wu et al., 2024; Zhang et al., 2024) evaluate MLLMs on synthesized images generated via carefully designed prompts using text-to-image models. While these methods can reveal general failure modes, they do not incorporate feedback from the MLLM itself, limiting their ability to uncover model-specific vulnerabilities or adapt flexibly across models. In contrast, GHOST leverages model feedback and performs targeted optimization in the CLIP image space.

DASH (Augustin et al., 2025) is the most comparable prior work, as it also incorporates MLLM feedback for image generation. However, it operates directly over diffusion latents and requires the MLLM, the diffusion model, the generated image, and an object detector to all remain in the optimization loop, making it both time and resource intensive and forcing the use of a distilled single-step diffusion model. Moreover, DASH ultimately retrieves similar images from the real dataset based on the generated ones. GHOST addresses these limitations with a decoupled and more efficient design. In Section 5 and Appendix C, we provide comparisons with DASH.

**Robustness of Diffusion Representations.** It is well-established that deep networks trained with empirical risk minimization (ERM) often rely on spurious correlations to make predictions (Sagawa et al., 2019; Kirichenko et al., 2022; Noohdani et al., 2024; Parast et al., 2025). Recent studies have also identified such correlations in CLIP (Radford et al., 2021b) and autoregressive vision-language models (Wang et al., 2024; Varma et al., 2024; Kim et al., 2023; Ye et al., 2024; Zheng et al., 2024). In contrast, diffusion models have shown greater robustness in their learned representations, as they aim to approximate the underlying data distribution rather than optimizing for a specific downstream task (Li et al., 2024; Luo et al., 2024). Li et al. (2024) demonstrate that U-Net-based latent diffusion models (Rombach et al., 2022) are less prone to shortcut solutions compared to discriminative models. DEEM (Luo et al., 2024) further argues that vision encoders in MLLMs, often based solely on models like CLIP-ViT, can be improved using the generative feedback of diffusion models to

align the semantic distributions of the image encoder. They showed this alignment enhances robustness to out-of-distribution inputs and reduces visual hallucinations. Motivated by these insights, we leverage diffusion models to probe vulnerabilities in MLLMs.

**Adversarial Attacks on MLLMs.** GHOST can be viewed as a form of adversarial generation, but we highlight key distinctions from prior work. Methods like AnyAttack (Zhang et al., 2025) and AttackVLM (Zhao et al., 2023) operate in pixel space and preserve the overall image semantics, aiming for imperceptible perturbations. In contrast, GHOST inserts semantic-level misleading cues, plausible to humans yet triggering hallucinations in MLLMs. While adversarial examples are sometimes transferable across models, GHOST exhibits a different kind of transfer: cues crafted for one model generalize to others, pointing to shared failure modes and spurious biases. Our approach is closer in spirit to content-based adversarial attacks such as Chen et al. (2023), but differs in both setting (image classifiers vs. MLLMs) and objective (causing hallucination rather than misclassification).

## 3 PROBLEM SETUP

We consider an MLLM $\mathcal{M}$ that integrates a vision encoder $\mathcal{V}_{\mathcal{M}}$ with a language model backbone $f_{\mathcal{M}}$. The text input is denoted as $\mathbf{X}_q \in \mathcal{L}$, where $\mathcal{L}$ is the space of language, and the image input as $\mathbf{X}_v \in \mathcal{I}$, where $\mathcal{I}$ is the space of images. The vision encoder $\mathcal{V}_{\mathcal{M}} : \mathcal{I} \to \mathbb{R}^{N \times d_{\mathcal{M}}}$ maps the image $\mathbf{X}_v$ into a sequence of $N$ vision tokens of dimension $d_{\mathcal{M}}$, denoted by $\mathbf{Z}_v = \mathcal{V}_{\mathcal{M}}(\mathbf{X}_v)$. These tokens, together with the text input $\mathbf{X}_q$, are passed to the LLM backbone $f_{\mathcal{M}}$, which generates an output sequence $Y = (y_1, \ldots, y_T)$ autoregressively. Each token $y_t$ is sampled from the conditional probability $p(y_t \mid y_{<t}, \mathbf{X}_q, \mathbf{Z}_v) = f_{\mathcal{M}}(y_{<t}, \mathbf{X}_q, \mathbf{Z}_v)$.

We also consider a Latent Diffusion Model $\mathcal{G}$, which generates an image $\tilde{\mathbf{X}}_v = \mathcal{G}(\mathbf{X}_v \mid c)$ given an initial image $\mathbf{X}_v$ and a conditioning signal $c$. We employ Stable Diffusion unCLIP (Rombach et al., 2022), where $c$ is an embedding in the CLIP embedding space. Formally, let $\mathcal{V}_{\text{CLIP}} : \mathcal{I} \cup \mathcal{L} \to \mathbb{R}^{d_{\text{CLIP}}}$ denote the CLIP encoder that learned joint representation for text and images. Note that $\mathcal{M}$ and $\mathcal{G}$ use different vision encoders; therefore, we introduce a mapper $\Pi : \mathbb{R}^{d_{\text{CLIP}}} \to \mathbb{R}^{N \times d_{\mathcal{M}}}$ to bridge these embedding spaces. Given an initial image $\mathbf{X}_v$ without the target object $t$, a prompt $\mathbf{X}_q$ of the form "Do you see a $t$ in the image?", and a target token $y^*$ (e.g.,"Yes"), we aim to find an embedding $c \in \mathbb{R}^{d_{\text{CLIP}}}$ that satisfies the following constraints:

1. **Proximity to the original image:** $c$ remains close to the initial embedding $c_0 = \mathcal{V}_{\text{CLIP}}(\mathbf{X}_v)$
2. **Absence of the target object semantics:** $c$ doesn't encode the target object itself.
3. **Inducing hallucination:** $\mathcal{M}$ assigns high probability to answering the target token $y^*$

$$p(y^* \mid \mathbf{X}_q, \Pi(c)) \geq \tau_{\text{yes}} \tag{1}$$

where $\tau_{\text{yes}}$ is a confidence threshold.

We condition the diffusion model on the optimized embedding $c$ to generate an image $\tilde{\mathbf{X}}_v = \mathcal{G}(\mathbf{X}_v \mid c)$ and check whether $\mathcal{M}$ hallucinates the target object. Diffusion unCLIP enables this process by allowing the image generation to be conditioned on a CLIP image embedding. Full pipeline and implementation details are provided in Section 4.

## 4 METHOD

This section introduces GHOST (**G**enerating **H**allucinations via **O**ptimizing **S**tealth **T**okens), a fully automated pipeline for stress-testing object hallucinations in MLLMs. We begin with the implementation of the mapper $\Pi$, then describe our optimization procedure, and finally explain how the diffusion model generates images within the pipeline.

### 4.1 BRIDGING EMBEDDING SPACES

To incorporate feedback from the MLLM into the generation process, a naive approach would require backpropagating through the entire pipeline – including the MLLM, the generated image, and the diffusion model– which is computationally expensive. To overcome this, we introduce a mapper

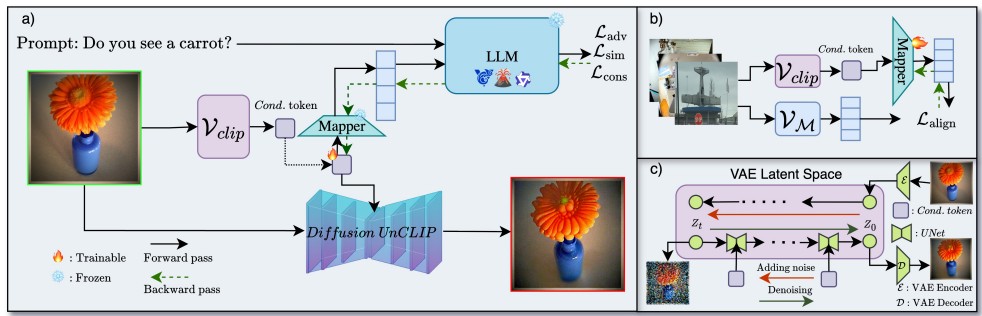

Figure 3: a) Overview of GHOST. We optimize only the CLIP embedding, then condition unCLIP on it, see (c) for decoding details. b) Training setup for the MLP, which aligns CLIP embeddings with the MLLM vision encoder using an MSE loss. c) A partially noised latent of the original image is denoised conditioned on the optimized embedding.

($\Pi$) that bridges the embedding spaces of the MLLM and the diffusion model, enabling more efficient optimization. We implement $\Pi$ as an MLP and find that this simple design is sufficient for our purposes. Given an MLLM $\mathcal{M}$ and a dataset $\mathcal{D}$ of images, the MLP is trained with a mean squared error (MSE) objective:

$$\mathcal{L}_{\text{align}} = ||\Pi(\mathcal{V}_{\text{CLIP}}(\mathbf{X}v)) - \mathcal{V}_{\mathcal{M}}(\mathbf{X}_v)||_2^2, \tag{2}$$

where $\mathbf{X}_v \in \mathcal{D}$. The training setup is illustrated in Figure 3b. For more details see Appendix D.

## 4.2 OPTIMIZATION OBJECTIVE

With the mapper $\Pi$ trained, we now introduce the attack. Given a target object $t$ and an initial image $\mathbf{X}_v$ that do not contain $t$, we optimize the CLIP embedding $z = \mathcal{V}_{\text{CLIP}}(\mathbf{X}_v)$ so that it tends to satisfy the conditions from Section 3.

**Proximity to the original image:** We regularize $c$ to remain close to the embedding of the initial real image. This prevents excessive drift and preserves high-level semantics. Formally, let $c_0$ be the CLIP embedding of the original image, we impose an $\ell_2$ penalty:

$$\mathcal{L}_{reg} = \left\| c - c_0 \right\|_2^2 \tag{3}$$

**Absence of the target object semantics:** To discourage $c$ from encoding the target object directly, we penalize its similarity to the CLIP embedding of the object's textual descriptions. Rather than relying on a single prompt, we define a set of textual templates $\mathcal{T}_{clip}$ (e.g., "a photo of a $t$") to represent the target object (See Appendix F.2 for more details). We compute the expected cosine similarity between $c$ and the CLIP embeddings of these templates:

$$\mathcal{L}_{\text{clip}} = \mathbb{E}_{\mathbf{T}_q \sim \mathcal{T}_{\text{clip}}} \left[ \cos(c, \mathcal{V}_{\text{CLIP}}(\mathbf{T}_q)) \right] \tag{4}$$

**Inducing hallucination:** To mislead the model to perceive the target object, we see the probability that the LLM assigns to the target token $y^*$. Specifically, given a query prompt $\mathbf{X}_q$ of the form "Do you see a $t$ in the image?", we define the loss as:

$$\mathcal{L}_{\text{adv}} = -\log p(y^\star \mid \mathbf{X}_q, \Pi(c)) \tag{5}$$

To prevent overfitting to a single phrasing, we construct a diverse set of semantically equivalent query templates and randomly sample $\mathbf{X}_q$ from this set at each optimization step (The templates are provided in AppendixF.4).

Our joint objective combines all components:

$$\mathcal{L}_{\text{total}} = \mathcal{L}_{\text{adv}} + \lambda_{clip}\,\mathcal{L}_{\text{clip}} + \lambda_{reg}\,\mathcal{L}_{\text{reg}}, \tag{6}$$

where $\lambda_{clip}$, $\lambda_{reg}$ are hyperparameters that control the relative importance of these terms. We minimize this loss using the AdamW (Loshchilov & Hutter, 2017) optimizer. An overview of this pipeline is illustrated in Figure 3a, and ablation studies are provided in Section 6.

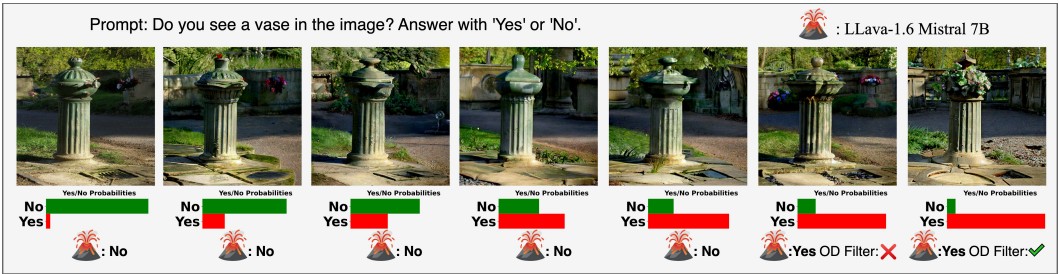

Figure 4: Optimization steps toward hallucination. We show the model's Yes/No probabilities for the optimized embedding at each step. Images are then generated by conditioning diffusion on that embedding. As "Yes" confidence increases, misleading cues (e.g., vase-like structures) emerge. Samples flagged by OWLv2 are discarded.

After each optimization step, we check whether the model's probability for the target token exceeds the threshold confidence $\tau_{yes}$. We perform at most $M$ optimization steps per image. If the probability does not surpass $\tau_{yes}$ within $M$ steps, we discard the result and move to the next image. Otherwise, once the threshold is met, we use the current embedding $z$ to generate a candidate image via the diffusion model, described in the next section.

### 4.3 GUIDED DIFFUSION

As shown in Figure 3c, instead of starting from pure noise, we begin the reverse process from a partially noised latent of the original image. This design encourages the final image to preserve the high-level structure of the input while still allowing for subtle semantic shifts that can induce hallucination. Concretely, we encode the original image $\mathbf{X}_v$ into its VAE latent representation, apply forward diffusion for $t$ steps to obtain $z_t$, and then condition the reverse denoising on $c$. The noise level $t$ controls the trade-off between preserving the original image structure and allowing space for misleading cues. To account for diffusion stochasticity, we allow up to $N = 4$ generation attempts per image. More details are provided in Appendix E.

We use an open-vocabulary object detector, OWLv2 (Minderer et al., 2024), as a filtering step to verify that the generated image does not actually contain the target object. If OWLv2 detects the target object, we conservatively discard the sample. Only when the model hallucinates the object and OWLv2 confirms its absence do we consider the hallucination-inducing process successful. Figure 4 provides a qualitative example: as we optimize the CLIP embedding to increase the model's confidence in answering "Yes," the images gradually incorporate misleading visual features, in this case, vase-like structures. Note that this figure is illustrative only; we do not generate images at each step of optimization. See Appendix I for additional detector analyses.

## 5 EXPERIMENTS

In this section, we present quantitative results of GHOST on three open-source MLLMs. We also evaluate the transferability of GHOST-generated images across a diverse set of additional models.

### 5.1 RESULTS

**Models.** We evaluate GHOST on three open-source MLLMs accessed via HuggingFace: Qwen2.5-VL-7B-Instruct (Bai et al., 2025), LLaVA-v1.6-Mistral-7B (Liu et al., 2024), and GLM-4.1V-Thinking (Team et al., 2025).

**Dataset.** We select 10 target object classes from the COCO dataset (Lin et al., 2014). For each class, we begin by excluding all images containing the target object based on COCO annotations. Among the remaining negatives, we select the top 1,000 images with the highest CLIP similarity to the object name. This sorting identifies visually diverse yet semantically related contexts where hallucination is more likely to occur. In Appendix F.1, we also evaluate GHOST without CLIP sorting and show that our results remain consistent. To isolate GHOST-induced failures, we discard any image where the MLLM already predicts the target object before optimization. We further evaluate GHOST

Table 1: **GHOST and DASH (Augustin et al., 2025) results on COCO.** "Samples" reflects the size of the input pool each method operates over.

| Method | Model | Samples | Hallucination | Success |
|--------|-------|---------|---------------|---------|
| GHOST | Qwen2.5-VL | 9423 | 2816 | 29.9% |
| | LLaVA-v1.6 | 8786 | 2468 | 28.1% |
| | GLM-4.1V | 8889 | 2880 | 32.4% |
| DASH-LLM | Qwen2.5-VL | 118,000 | 57 | 0.1% |
| | LLaVA-v1.6 | 118,000 | 153 | 0.1% |
| DASH-OPT | Qwen2.5-VL | 118,000 | 42 | 0.1% |

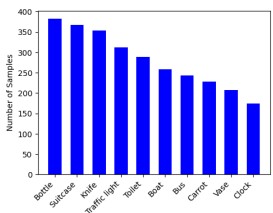

Figure 5: Class-wise hallucination samples generated by GHOST across 10 objects for Qwen.

on ObjectNet (Barbu et al., 2019) to test its behavior on a different visual distribution; results are provided in Appendix G.

**Results.** We present the GHOST results in Table 1. For each model, "Samples" refers to the number of images (out of the initial 10,000 COCO images) in which the model does not perceive the target object. We then report how many hallucination-inducing images GHOST successfully generates, followed by the success rate (i.e., number of hallucinations divided by the number of samples). We also include results from DASH (Augustin et al., 2025), applied to the COCO dataset. Despite operating on a much smaller image pool, GHOST discovers orders of magnitude more hallucination-inducing samples. For instance, GHOST identifies 2,816 successful cases for Qwen2.5-VL, while DASH-LLM and DASH-OPT combined find only 99. It is important to note that DASH was originally designed to search over ReLAION-5B (LAION, 2024), a massive web-scale dataset. Its goal is not to maximize hallucination count, but to uncover naturally occurring failure cases across large-scale image corpora. Our comparison on COCO is intended to provide a point of reference against an existing method that shares a similar objective: identifying systematic hallucination-inducing inputs. Further details about our DASH experiments are provided in Appendix C. We also break down GHOST's performance by object class in Figure 5. GHOST discovers a substantial number of hallucination cases for all classes. More details on the class-wise results of our method are provided in Appendix K.

**Image Quality.** To evaluate the quality of the samples generated by GHOST, we conduct both qualitative and quantitative analyses. Figure 6 presents visual comparisons between GHOST and two baseline diffusion methods: Stable Diffusion v2.1 (Rombach et al., 2022), where we prompt the model using the COCO caption of the image, and Stable Diffusion unCLIP, where the diffusion is conditioned on the CLIP embedding of the image. GHOST preserves the semantics of the original image while introducing minimal visual artifacts. Table 2 reports Fréchet Inception Distance, FID scores (Heusel et al., 2017), evaluating each method along two axes: (i) realism, measured by FID against COCO validation images, and (ii) semantic fidelity, measured by FID against the initial images used in diffusion. While GHOST achieves comparable realism to baseline methods, it outperforms them in semantic preservation, confirming its advantage in maintaining visual identity. We observe higher FID scores for GLM hallucinations, likely due to the model's reasoning nature. Since we optimize a "thinking" token rather than the final answer directly, the optimization path is less aligned with the hallucination objective, making it harder to induce consistent visual drift. We further report SSIM (Wang et al., 2004) perceptual similarity results in Appendix F.9. Also more qualitative samples are provided in Appendix K.

**Reasoning Model.** We evaluate GHOST on GLM-4.1V-Thinking (Team et al., 2025), a multimodal model trained with a reasoning-centric framework. The model structures its responses in two stages: it first generates a reasoning trace enclosed within `<think>`...`</think>` tokens, followed by a final answer within `<answer>`...`</answer>`. To adapt GHOST to this format, we use the probability of the token `"Yes"` at the first decoding step after the `<think>` token. This allows us to keep the optimization objective and runtime consistent with other models. We find that, despite not explicitly optimizing for the final answer, this is sufficient to induce hallucinated `"Yes"` responses in the final output. A qualitative example is shown in Figure 7, where GHOST edits shift the model's reasoning trajectory to justify the presence of a nonexistent object.

Original   GHOST

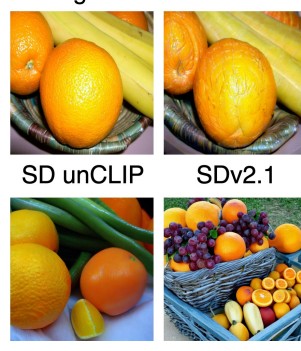

SD unCLIP  SDv2.1

Figure 6: Qualitative comparison of generated images.

Table 2: FID scores (lower is better) for different diffusion methods. Each method is evaluated for realism (vs. COCO validation images) and semantic preservation (vs. the initial images used in diffusion).

| Method | Qwen2.5-VL | LLaVA-v1.6 | GLM4.1V |
|---|---|---|---|
| *Distributional Realism* | | | |
| SD v2.1 | **46.19** | **48.42** | 44.79 |
| SD unCLIP | 46.51 | 50.20 | **44.76** |
| GHOST | 47.03 | 50.78 | 51.70 |
| *Semantic Fidelity* | | | |
| SD v2.1 | 41.71 | 42.64 | 39.85 |
| SD unCLIP | 31.67 | 35.47 | **32.07** |
| GHOST | **25.00** | **26.39** | 34.94 |

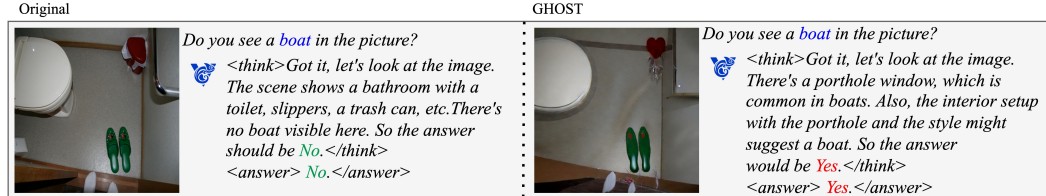

Figure 7: Applying GHOST to a reasoning model (GLM-4.1V-Thinking).

**Human Evaluation.** To evaluate whether GHOST images preserve the absence of the target object from a human perspective, we ask 40 peers to review the images, resulting in over 3,000 total votes. They see the final GHOST images and indicate whether they can see the target object. On average, 89% of responses for images optimized w.r.t LLaVA-v1.6 and 86.3% for Qwen2.5-VL indicate that the object is not present. For reference, DASH (Augustin et al., 2025) reports that 5.2% of their images contained the object and 7.8% were ambiguous, implying that 87% of their samples are perceived as object-free. We also evaluate the naturalness of GHOST images and find that they are rated comparably natural to the diffusion baseline. Additional details are provided in Appendix B.

**Runtime Efficiency.** GHOST requires ~10 seconds per sample on a single A100 GPU, making it about $5\times$ faster than DASH on average, even when DASH uses a distilled diffusion model. Detailed runtimes are reported in Appendix F.8. We also report runtime measurements at higher image resolutions in Appendix F.10.

## 5.2 Transferability

We evaluate how images generated by GHOST on one model transfer to others. In addition to the three source models, we include two larger open-source models: LLaMA3.2 11B (Meta, 2024) and Aya 32B (Dash et al., 2025), as well as two closed-source models: GPT-4o (OpenAI, 2024) and Gemini2.5-Flash (Comanici et al., 2025). As shown in Table 3, images optimized on Qwen2.5-VL achieve a 66.5% hallucination success rate on GPT-4o. These results suggest that different models share common failure patterns. More details are provided in Appendix K.

## 6 Ablation Studies

### 6.1 Effect of Hyperparameters

**Effect of $\tau$.** We study how the confidence threshold $\tau$ impacts GHOST's success by varying $\tau \in \{0.5, 0.6, 0.7, 0.8, 0.9\}$ on 400 images from two objects (200 images each), while keeping the number of optimization steps fixed at 100 and using the same learning rate. A higher $\tau$ re-

Table 3: Transferability of hallucinations across models. Each row corresponds to the model used, and each column shows the success rate (%) on a target model.

| Model | Qwen2.5-VL | LLaVA-v1.6 | GLM4.1V | GPT-4o | Aya | LLaMA3.2 | Gemini |
|---|---|---|---|---|---|---|---|
| Qwen2.5-VL | – | 62.2 | 72.0 | 66.5 | 71.1 | 65.8 | 58.6 |
| LLaVA-v1.6 | 52.6 | – | 50.5 | 50.5 | 54.4 | 49.7 | 42.8 |
| GLM4.1V | 63.2 | 57.1 | – | 63.8 | 67.6 | 69.1 | 53.8 |

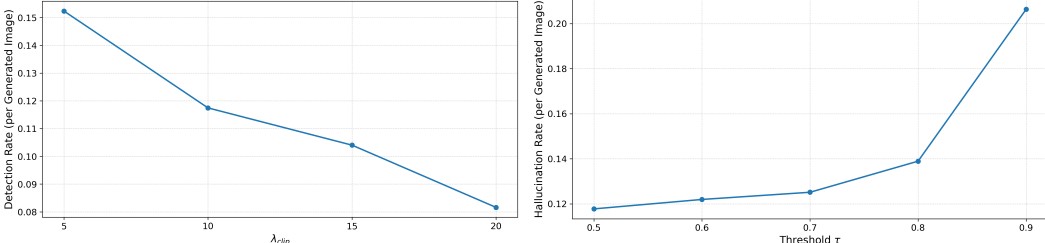

Figure 8: Larger $\lambda_{\text{clip}}$ leads to less objects.      Figure 9: Larger $\tau$ leads to more hallucination.

quires the optimized embedding to achieve higher $p(y^* \mid \mathbf{X}_q, \Pi(c))$, making optimization harder. As expected, this reduces the number of images that meet the threshold within the allowed steps (see Appendix F.7). However, as shown in Figure 9, images generated from embeddings optimized with higher $\tau$ are more likely to induce hallucination. These results confirm that the confidence score $p(y^* \mid \mathbf{X}_q, \Pi(c))$ serves as a meaningful proxy for the model's belief in object presence, even though the optimization is decoupled from the diffusion process.

**Effect of $\lambda_{\text{clip}}$.** We evaluate the impact of $\lambda_{\text{clip}}$ using the same setup. As shown in Figure 8, increasing $\lambda_{\text{clip}}$ reduces the object detection rate per generated image. This confirms that stronger penalties on CLIP similarity discourage the diffusion model from generating images containing the object. Additional results are provided in Appendix F.6.

**Effect of $\lambda_{reg}$.** As shown in Appendix F.5, increasing $\lambda_{reg} \in \{1.0, 1.5, 2.0\}$ on GLM samples lowers FID scores, indicating that this term helps preserve visual realism.

## 6.2 MITIGATION

As a small toy setup, we fine-tune Qwen2.5-VL with LoRA (Hu et al., 2022) using GHOST images. Following DASH (Augustin et al., 2025), we include both positive and negative samples to avoid forgetting of object concepts. However, unlike DASH, which uses real COCO images as positives, we generate both positives and negatives synthetically using Diffusion unCLIP to ensure consistency in visual style. This prevents the model from simply learning to reject all synthetic content.

Table 4 shows that fine-tuning Qwen2.5-VL on GHOST images improves robustness to hallucination. We observe consistent gains on POPE (Li et al., 2023) as well as on the CHAIR benchmark, following the evaluation setup from Li et al. (2023). Also, performance on general vision-language tasks (VQAv2 and COCO Captioning) remains nearly unchanged, suggesting that GHOST-based fine-tuning preserves general capabilities. Additional fine-tuning details can be found in Appendix J.

Table 4: Fine-tuning Qwen2.5-VL on GHOST images improves robustness to hallucination.

| | POPE ↑ | CHAIRs ↓ | CHAIRi ↓ | VQAv2 ↑ | Caption ↑ |
|---|---|---|---|---|---|
| Baseline | 88.7 | 3.8 | 3 | **89.5** | **72.8** |
| Finetuned | **93.2** | **2.9** | **2.6** | 89.4 | 71.5 |

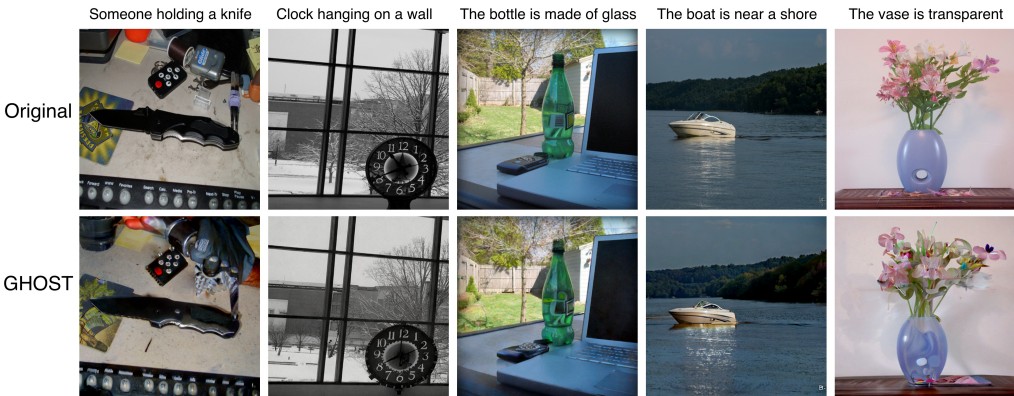

Figure 10: Applying GHOST to other types of hallucination.

## 6.3 Extending GHOST

GHOST primarily targets object hallucinations. However, to demonstrate its broader potential, we conduct a small experiment to test whether GHOST can also stress-test other types of hallucinations, such as attribute or relation hallucinations. Using the same losses as in the main setting, we select samples for which the victim model originally answers "No" to five queries, including "someone is holding a knife," "the vase is transparent," "the bottle is made of glass," "the clock is hanging on the wall," and the relation query "the boat is near the shore." We then optimize the embedding to flip the first-token prediction to "Yes." No object detector is used in this setting, as these queries do not correspond to object presence. Qualitative samples are shown in Figure 10. Across these five attribute/relation queries, GHOST achieves an average success rate of 32.2%. More details are provided in Appendix H.

## 7 Limitation & Future work

A key practical limitation of GHOST is its reliance on diffusion models that support image embedding-level conditioning, such as unCLIP. GHOST optimizes directly in the vision-embedding space, which requires a generative model capable of decoding modified embeddings back into pixel space. Many popular diffusion architectures do not currently provide this interface and therefore cannot serve as drop-in replacements in our framework. In addition, our current evaluation primarily focuses on object-centric hallucinations; a more systematic investigation across a broader range of hallucination modalities, along with more extensive corrective evaluations across architectures and hallucination types, would provide a more comprehensive assessment.

## 8 Conclusion

We introduce GHOST, a pipeline for generating images that induce hallucinations in MLLMs by inserting misleading cues. Our method leverages diffusion models to craft these cues, demonstrating that their vision space aligns well with the visual representations used by MLLMs. Beyond a single model, GHOST reveals cross-modal failure modes. We hope GHOST can help and inspire future work toward building more robust and reliable multimodal systems.

**Reproducibility.** We provide detailed descriptions of all components to facilitate reproducibility. Appendix D outlines the training procedure for the mapper, Appendix F describes the GHOST pipeline, and Appendix J covers the fine-tuning setup and evaluation. We have also submitted our code as supplementary material.

## ACKNOWLEDGMENTS

Naveed Akhtar is a recipient of the Australian Research Council Discovery Early Career Researcher Award (project # DE230101058) funded by the Australian Government. This work is also partially supported by Google Research Scholar Program Award. This research was supported by The University of Melbourne's Research Computing Services and the Petascale Campus Initiative. Aryan Yazdan Parast was partly supported by the Rowden White Scholarship (The University of Melbourne). This project was also supported in part by a grant from an NSF CAREER AWARD 1942230, the ONR PECASE grant N00014-25-1-2378, ARO's Early Career Program Award 310902-00001, Army Grant No. W911NF2120076, the NSF award CCF2212458, NSF Award No. 2229885 (NSF Institute for Trustworthy AI in Law and Society, TRAILS), a MURI grant 14262683, DARPA AIQ grant HR00112590066 and an award from meta 314593-00001.

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

## A  AI Usage Clarification

Large Language Models were used to to aid with writing clarity and polish.

## B  Human Evaluation

To evaluate whether GHOST-generated images preserve the absence of the target object from a human perspective, we design a simple interface and collect annotations from peers. For both LLaVA and Qwen2.5-VL, we randomly sample 50 successful GHOST images per object across 10 object categories. To validate annotator reliability, we include control images that are known to contain the target object. These control images are generated using the same Diffusion unCLIP model, but conditioned on positive samples, ensuring the object is clearly present. Examples are shown in Figure 17. We include 50 control images per object.

Each annotator is shown a set of approximately 100 randomly selected images (comprising 20% control images, 40% GHOST–Qwen, and 40% GHOST–LLaVA) and asked "Is there a [target object] in this image?" with binary response options: "Yes" or "No." Before beginning the evaluation, annotators completed a brief training phase (Figure 11) that displayed a few labeled examples to calibrate their expectations.

In total, we collect 1,590 votes on LLaVA images, 1,607 on Qwen images, and 1,056 on control images, from 40 unique participants. Aggregate results are shown in Table 5, reporting the percentage of "Yes" responses in each group. Figure 12 further breaks down the human yes-rate by object class. We used the human evaluation setup from Moakhar et al. (2024).

Table 5: Human yes rate on control and GHOST-generated images.

| Set of Images: | Control Images | GHOST–LLaVA | GHOST–Qwen |
|---|---|---|---|
| **Human Yes Rate** | 91.2% | 11.0% | 13.7% |

**Naturalness**. The goal of GHOST is not to produce perfectly photorealistic images, but to create semantically meaningful perturbations that reveal model vulnerabilities. While FID scores in the main paper show that GHOST preserves the quality of the underlying diffusion model, we additionally assess naturalness with a human study. We sample 30 GHOST images and 30 unCLIP-generated counterparts guided by the same initial images. Volunteers rate naturalness on a 1–5 scale (1: completely unnatural, 5: completely natural). As shown in Table 6, GHOST images receive ratings comparable to the diffusion baseline (3.55 vs. 3.56 on average), with a similar proportion rated "mostly natural" (ratings $\geq 4$).

## C  DASH

We reproduced DASH results with Qwen2.5 and Llava. For Llava we only investigated DASH-LLM which is a varient of DASH results. We omited DASH-OPT since even batch size of 1 could not be achieved with 80GB gpus. For Qwen2.5 we investigated both DASH-LLM and DASH-OPT. For DASH-OPT we finetuned three parameters, step size, gradient clip and vlm weight on the loss. for step size and gradient clip we investigated $[0.1, 0.2, 0.4, 0.8]$ and for MLLM weight we checked $[1, 2, 4, 6, 8]$. Our meassure was the number images that does not contain the object according to object detector but mislead the MLLM. We could not use ReLION-5B as the KNN search service is private. In addition calculating clip embeddings on all those images was not practical for us. We used COCO dataset instead. On COCO these are the results.

## D  Mapper

### D.1  Mapper Training

To bridge the embedding spaces, we introduce a mapper $\Pi$ that projects the CLS token from the CLIP vision encoder space $\mathcal{V}_{\text{CLIP}}$ into the vision encoder space of the target multimodal model. We

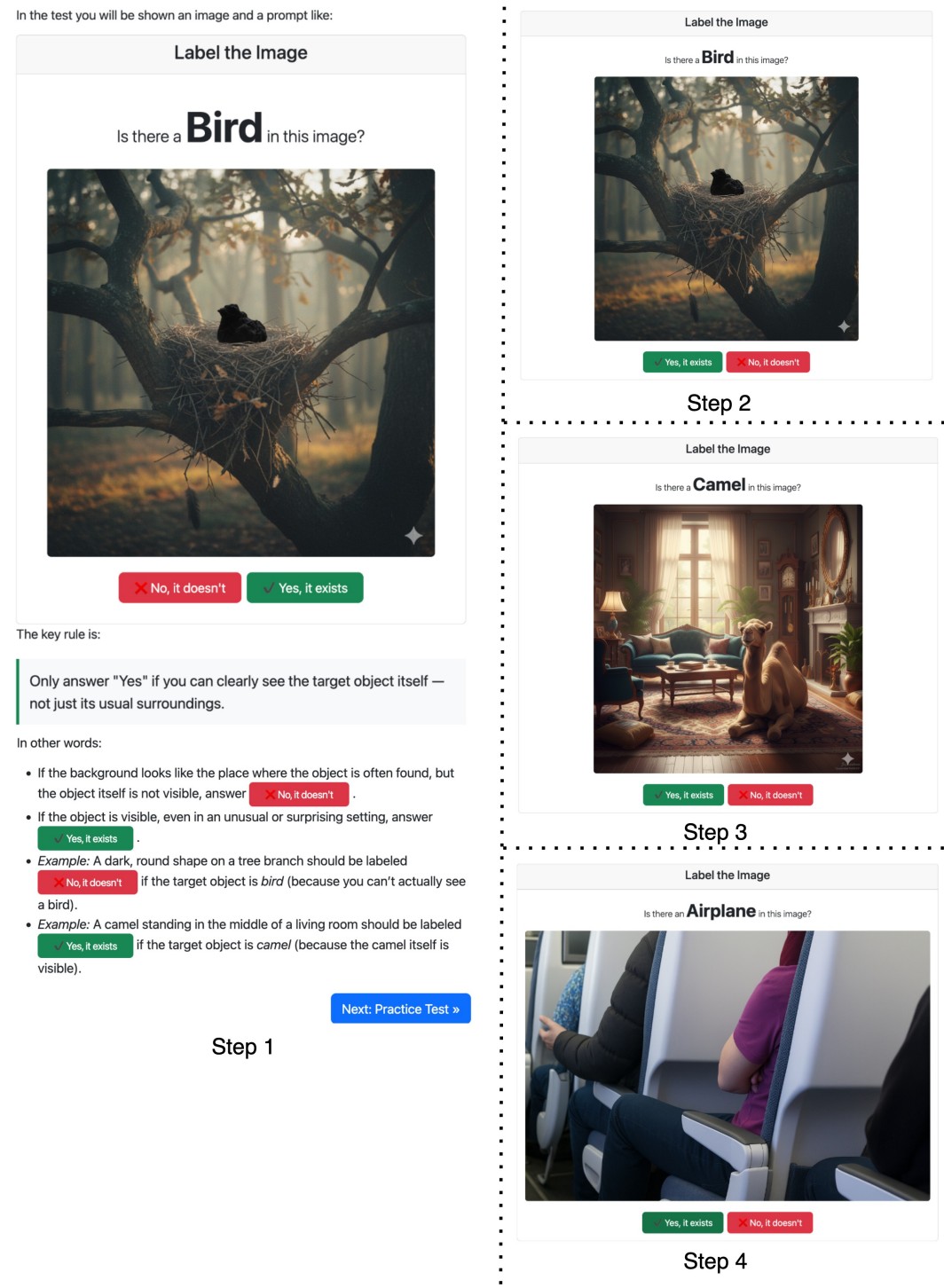

Figure 11: Human Evaluation Training Phase

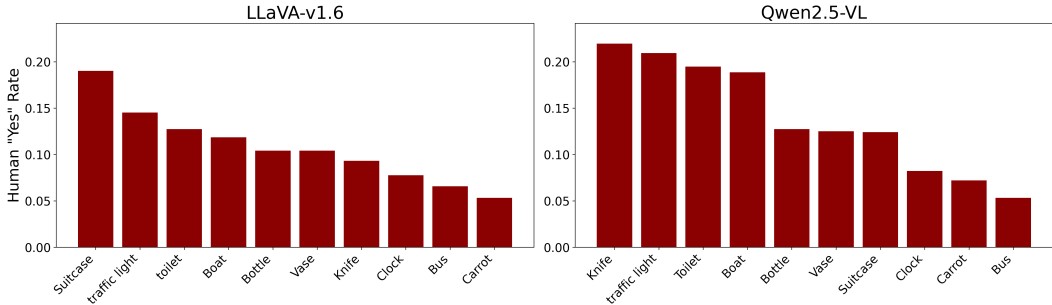

Figure 12: Human Yes-Rate by object class

Table 6: Human Evaluation for Naturalness of GHOST Images.

| Model | Avg. Rating | # Votes | # $\geq$ 4 |
|---|---|---|---|
| GHOST | 3.55 | 191 | 111 |
| SD-unCLIP | 3.56 | 189 | 118 |

Table 7: The number of successful image found by DASH on COCO

| Method | Number of Found Images |
|---|---|
| DASH-opt (qwen2.5) | 42 |
| DASH-LLM (qwen2.5) | 57 |
| DASH-LLM (llava) | 153 |

design a simple multi-layer perceptron (MLP) to achieve this mapping. Specifically, the mapper takes the CLIP CLS token $\mathbf{z}_{\text{CLIP}} \in \mathbb{R}^{d_{\text{CLIP}}}$ as input and outputs a sequence of $N$ tokens, each of dimension $d_{\mathcal{M}}$, suitable for the target model:

$$\Pi : \mathbb{R}^{d_{\text{CLIP}}} \rightarrow \mathbb{R}^{N \times d_{\mathcal{M}}}.$$

Concretely, we first broadcast the CLS token across $N$ positions, obtaining

$$\mathbf{Z}_{\text{rep}} = [\mathbf{z}_{\text{CLIP}}, \dots, \mathbf{z}_{\text{CLIP}}] \in \mathbb{R}^{N \times d_{\text{CLIP}}}.$$

In parallel, we maintain a set of learnable context tokens

$$\mathbf{E} = \{\mathbf{e}_1, \dots, \mathbf{e}_N\}, \quad \mathbf{e}_i \in \mathbb{R}^{d_{\mathcal{M}}},$$

which provide token-specific priors for the mapping. For each position $i$, we concatenate the repeated CLS token with its corresponding learnable context token:

$$\mathbf{h}_i = [\mathbf{z}_{\text{CLIP}}; \mathbf{e}_i] \in \mathbb{R}^{d_{\text{CLIP}} + d_{\mathcal{M}}}.$$

The sequence $\mathbf{H} = \{\mathbf{h}_1, \dots, \mathbf{h}_N\}$ is then passed through an MLP with two hidden layers of size $d_h$, each followed by a GELU activation:

$$\mathbf{y}_i = \text{MLP}(\mathbf{h}_i), \quad \mathbf{Y} = \{\mathbf{y}_1, \dots, \mathbf{y}_N\} \in \mathbb{R}^{N \times d_{\mathcal{M}}}.$$

The learnable context embeddings $\mathbf{E}$ serve as token-specific conditioning vectors. Although the MLP parameters are shared across all positions, the concatenation $[\mathbf{z}_{\text{CLIP}}; \mathbf{e}_i]$ ensures that each target token $i$ is shaped by both the global CLIP semantics and its unique context embedding. For training the mapper, we used the AdamW optimizer with cosine annealing learning rate scheduling. The details of the training process and hyperparameters are summarized in Table 11.

Table 8: Number of successful image found by DASH on COCO by class name

| Class/Method | DASH-opt (qwen2.5) | DASH-LLM (qwen2.5) | DASH-LLM (llava) |
|---|---|---|---|
| traffic light | 0 | 0 | 0 |
| carrot | 1 | 0 | 10 |
| knife | 0 | 2 | 14 |
| clock | 13 | 11 | 7 |
| toilet | 0 | 0 | 0 |
| boat | 2 | 0 | 0 |
| suitcase | 3 | 1 | 11 |
| bottle | 20 | 42 | 80 |
| vase | 3 | 0 | 23 |
| bus | 0 | 1 | 8 |

Table 9: Successful images found by DASH on COCO by class name

| Class/Method | DASH-opt (qwen2.5) | DASH-LLM (qwen2.5) | DASH-LLM (llava) |
|---|---|---|---|
| clock |  |  |  |
| suitcase |  |  |  |
| bottle |  |  |  |

Table 10: Ablation on hidden dimension and context dimension for the projector, evaluated on LLaVA and Qwen. Numbers report accuracy (%). The final configuration selected for our experiments is highlighted in **bold**.

|       | Hidden Dim     | 512   | 1024    | 2048    | 4096  |
|-------|----------------|-------|---------|---------|-------|
| LLaVA | Context = 6144 | 70.17 | 72.17   | 72.75   | 73.67 |
|       | Context = 4096 | 70.83 | **72.25** | 72.50 | 72.67 |
| Qwen  | Context = 6144 | 69.50 | 71.08   | 73.50   | 73.92 |
|       | Context = 4096 | 68.08 | 71.67   | **73.92** | 72.75 |

Table 11: Training hyperparameters for each model.

| Model | LR   | Epochs | Batch Size | Weight Decay | Scheduler                   | Warmup Steps |
|-------|------|--------|------------|--------------|-----------------------------|--------------|
| GLM   | 2e-4 | 10     | 32         | 0.01         | Cosine ($T_{\max} = 10$)    | 1000         |
| LLaVA | 2e-4 | 10     | 64         | 0.01         | Cosine ($T_{\max} = 10$)    | 1000         |
| Qwen  | 2e-4 | 10     | 32         | 0.01         | Cosine ($T_{\max} = 10$)    | 1000         |

## D.2 MODEL SELECTION.

For model selection, we randomly sampled 100 images containing a given object class and 100 images without it from the COCO training set. We then evaluated the accuracy of the MLLM using the bridged embeddings obtained from the mapper $\Pi$, and the same prompts used in the main method (e.g. Do you see the object in the image). We considered six object classes: *vase*, *boat*, *bird*, *giraffe*, *car*. These classes were selected to cover a range of object characteristics: small objects such as *remote*, which require fine-grained detail to be detected from the mapper, and large objects such as *giraffe*, which are comparatively easier for the MLLM to recognize. In Table 10, we report the accuracy obtained with different settings of context dimension and hidden dimension. The final configuration selected for our experiments is highlighted in **bold**. We chose this setting by balancing accuracy with model size and efficiency, ensuring that the mapper achieves strong performance without incurring unnecessary computational cost.

## D.3 MAPPER EVALUATION

Inspired by (Liu et al., 2023a), we assess the reconstruction capability of the MLP connector $f$ using GPT-4 as a judge. For each input image $x$, we first obtain the response of the target MLLM (e.g., LLaVA) when conditioned on the true image embedding $\mathcal{M}(\mathcal{E}_v(x))$. We then provide this response, together with the ground-truth COCO object annotations, to GPT-4, which rates the consistency between the response and the annotations. Next, we replace the image input with the projected embedding $f(\mathcal{E}_{\text{clip}}(x))$, obtain the MLLM response, and request GPT-4 to rate it in the same way. The relative score between the two ratings serves as a measure of the mapper's ability to reconstruct the semantics of the image. We define the relative score as the ratio between the GPT rating of the projected-embedding response and that of the real-image response which quantifies how much semantic information from the original image embedding is retained after projection. Further details on this experiment and the evaluation prompt are provided in the appendix. For this study, we randomly selected 100 images from the COCO dataset. The results on LLaVA-v1.6, Qwen2.5-VL, and GLM are reported in Table 12. As shown, the projector preserves most of the semantic content of the original image while substantially reducing computational cost.

## E LATENT DIFFUSION MODELS

Diffusion probabilistic models (Sohl-Dickstein et al., 2015; Ho et al., 2020) generate samples that approximate the data distribution through a parameterized Markov chain. The core idea is to gradually corrupt a clean sample with Gaussian noise (forward process), and then learn to reverse this process (backward process) to recover clean data.

Table 12: Evaluation of projector reconstruction using GPT-4 as judge. Scores are reported for MLLM responses conditioned on the real image and on the reconstructed embedding.

| Source Model | Input Type | Basic Score | Relative Score |
|---|---|---|---|
| Qwen2.5-VL | Real Images | 89.6 | 76.0 |
| | Reconstructed | 68.1 | |
| LLaVA-v1.6 | Real Images | 84.9 | 54.5 |
| | Reconstructed | 46.3 | |
| GLM4.1-Thinking | Real Images | 94.0 | 75.6 |
| | Reconstructed | 71.1 | |

Latent Diffusion Models (LDMs) (Rombach et al., 2022) perform this procedure in a compressed latent space. Let $z_0$ denote the latent representation of a clean image obtained via a VAE encoder ($\mathcal{E}_{vae}$.) The forward noising process defines a distribution

$$q(z_t \mid z_0) = \mathcal{N}\big(z_t; \sqrt{\bar{\alpha}_t}\, z_0, \, (1 - \bar{\alpha}_t)I\big),$$

where $t \in \{1, \ldots, T\}$ is the diffusion timestep, $\alpha_t$ are variance schedule parameters, and $\bar{\alpha}_t = \prod_{s=1}^{t} \alpha_s$. Thus, a latent $z_t$ can be obtained directly by applying the scheduler to $z_0$, progressively injecting noise as $t$ increases.

The reverse process learns to iteratively denoise, starting from $z_T \sim \mathcal{N}(0, I)$ or, more generally, from a noisy latent $z_t$:

$$z_{t-1} = \frac{1}{\sqrt{\alpha_t}} \left( z_t - \frac{1 - \alpha_t}{\sqrt{1 - \bar{\alpha}_t}} \, \epsilon_\theta(z_t, t, C) \right) + \sigma_t \mathbf{z},$$

where $\epsilon_\theta$ is a UNet denoiser trained on large-scale data, $C$ is an optional conditioning signal (e.g., text or image embeddings), and $\mathbf{z} \sim \mathcal{N}(0, I)$. Finally, the clean latent $z_0$ is decoded through the VAE decoder $\mathcal{D}$ to obtain the generated sample in pixel space.

In this work, we employ the Stable Diffusion unCLIP model (Rombach et al., 2022), which conditions the denoising process on CLIP image embeddings (Radford et al., 2021a). Given an image $x$ and a CLIP image encoder $\mathcal{E}_{\text{clip}}$, the model generates a new sample conditioned on $\mathcal{E}_{\text{clip}}(x)$, ensuring that the output preserves the high-level semantics of $x$.

# F   GHOST OPTIMIZATION

## F.1   GHOST WITHOUT CLIP SORTING

For evaluating the performance of GHOST without CLIP-based sorting, we conducted an experiment on four classes: "Boat", "Vase", "Knife", and "Bottle". For each class, we randomly selected 200 samples that do not have the target object from their categories based on COCO annotations. When we select the pictures without sorting, the initial images are farther from the target object semantically, so more changes are needed to induce hallucination. To overcome this issue, we used noise level $t = 15$ (in contrast to our standard setting where the noise level is $t = 30$) as a hyperparameter to add more noise to the image and be able to change the image further. We also selected $\tau = 0.6$ to soften the constraint of optimization; all the other hyperparameters are the same as the standard setting. As reported in Table 13, the success rate is comparable to our method in the sorting setting. Qualitative samples with $t = 15$ as the noise level are also shown in Figure 13. As shown in Figure 13, when the noise level is decreased, more changes are applied to the initial images to induce hallucination.

## F.2   TEXT REPRESENTATION

The $\mathcal{L}_{\text{clip}}$ term in our objective function (Equation 4) prevents the optimized image embedding $z$ from directly encoding the semantics of the target object. To make this regularization robust, we use an enhanced, compositional text representation of the target object, rather than a single word

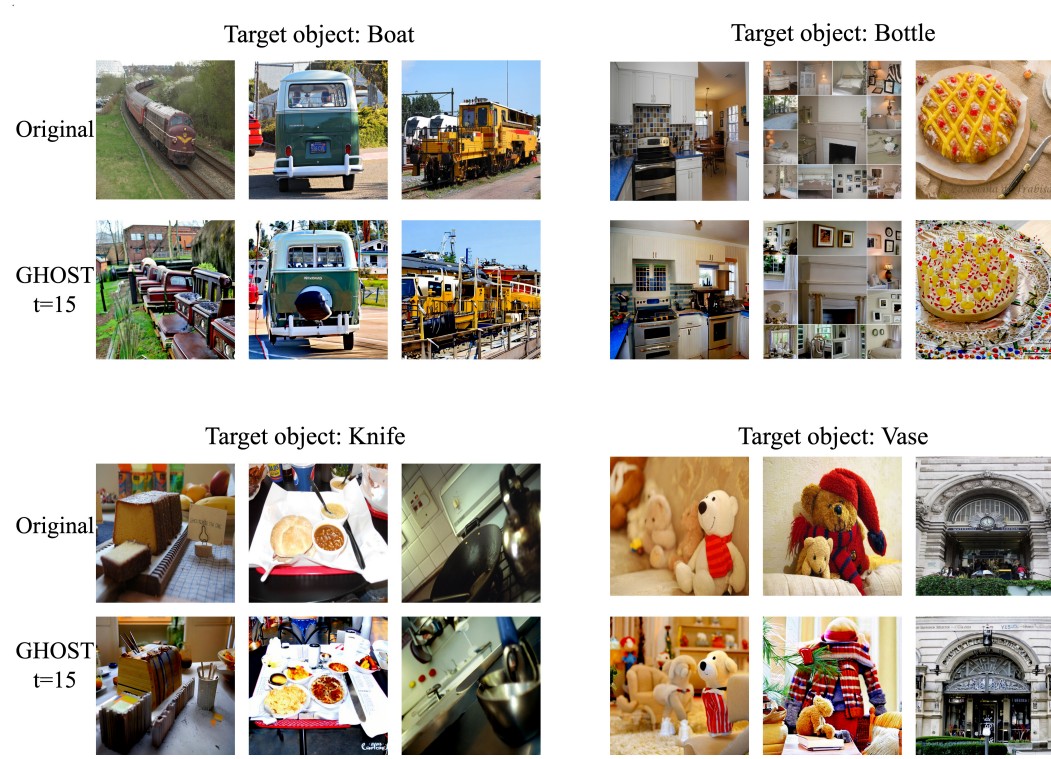

Figure 13: Qualitative samples from GHOST on Qwen2.5 VL 7B, with noise level $t = 15$. The top row shows the original images, and the bottom row shows samples generated with GHOST.

Table 13: Class-wise hallucination results at noise level $t = 15$.

| Class | Samples | Hallucination | Success (%) |
|---|---|---|---|
| Vase | 198 | 63 | 32% |
| Knife | 198 | 91 | 46% |
| Boat | 200 | 48 | 24% |
| Bottle | 188 | 107 | 57% |
| **Total / Mean** | 784 | 309 | 40% |

```
generic_templates = [
    "A scene featuring a {class_name}",
    "An image showing a {class_name}",
    "A photograph with a {class_name}",
    "A picture containing a {class_name}"
]
```

Listing 1: Generic contextual templates used for constructing the enhanced text representation. The placeholder 'class_name' is replaced with the target object name during embedding generation.

textual description. This representation helps push the optimized image embedding away from a comprehensive semantic understanding of the object, encouraging the MLLM to hallucinate based on subtle visual cues instead of object generation.

Our compositional text representation for a target object $t$ is constructed from three distinct sources, whose CLIP embeddings are then combined via a weighted average:

1. **Direct Object Description:** We start with a straightforward descriptive sentence, such as "A photo of a *class_name*". This provides a simple, unambiguous representation of the object. Let its CLIP embedding be $E_D$.

2. **Generic Contextual Templates:** We utilize a set of generic templates to capture various linguistic contexts. These templates are universal across all target objects and are listed in Listing 1. For a given target object $t$, these templates are filled to form specific phrases (e.g., "A scene featuring a vase"). Up to $N_{GT} = 4$ such template-based phrases are selected, and their CLIP embeddings are denoted as $E_{GT,j}$ for $j = 1, \dots, N_{GT}$.

3. **Mined COCO Captions:** We augment the representation by identifying and embedding actual captions from the COCO training dataset that explicitly contain the target object. Incorporating real-world descriptive language captures nuanced ways humans refer to objects in visual contexts. Up to $N_{CC} = 5$ such mined captions are selected, and their CLIP embeddings are denoted as $E_{CC,k}$ for $k = 1, \dots, N_{CC}$.

These CLIP text embeddings are combined into a single 'compositional embedding' ($E_{\text{comp}}$) using a weighted average. The formula is:

$$E_{\text{comp}} = w_D E_D + \sum_{j=1}^{N_{GT}} \frac{w_{GT}}{N_{GT}} E_{GT,j} + \sum_{k=1}^{N_{CC}} \frac{w_{CC}}{N_{CC}} E_{CC,k} \tag{7}$$

where $w_D = 0.3$, $w_{GT} = 0.4$, and $w_{CC} = 0.3$ are the base weights for the direct description, generic templates, and COCO captions, respectively. The weights for generic templates and COCO captions are distributed evenly among their respective constituent embeddings. This robust $E_{\text{comp}}$ is then used as the specific text query $\mathbf{T}_q$ for the $\mathcal{L}_{\text{clip}}$ loss term, defined as $\cos(z, \mathcal{V}_{\text{CLIP}}(E_{\text{comp}}))$. By minimizing this loss, we encourage $z$ to be *dissimilar* to $E_{\text{comp}}$, thereby preventing the optimized image embedding from inadvertently encoding the actual presence of the target object.

### F.3 HYPERPARAMETERS

We summarize the attack hyperparameters for each victim model in Table 14, while the corresponding model identifiers are listed separately in Table 15. We tuned the hyperparameters based on our evaluation metrics, including image quality (FID score), attack success rate, and human evaluation, which all are discussed in the main paper.

### F.4 PROMPTS

At each step of the optimization, we randomly select a prompt from our template set to avoid overfitting to a specific prompt. The prompts are designed to query the presence of the target object in a binary (Yes/No) format. This randomization ensures that the optimization does not exploit superficial linguistic patterns but instead focuses on inducing the desired hallucination. The complete set of prompt templates is provided in Listing 2.

Table 14: Attack hyperparameters for each victim model.

| Model | lr | total steps | $\tau$. | N | Guidance Scale | $\lambda_{clip}$ | $\lambda_{reg}$ | OD thr. | t | num_inf. |
|-------|-----|-------------|---------|---|----------------|------------------|-----------------|---------|-----|----------|
| Qwen | 0.1 | 100 | 0.8 | 4 | 5.0 | 15.0 | 10.0 | 0.5 | 30 | 50 |
| LLaVA | 0.1 | 100 | 0.8 | 4 | 5.0 | 15.0 | 10.0 | 0.5 | 30 | 50 |
| GLM | 0.2 | 125 | 0.5 | 5 | 5.0 | 0.5 | 1.5 | 0.5 | 30 | 50 |

Table 15: Model identifiers.

| Model | ID / Name |
|-------|-----------|
| Qwen | Qwen/Qwen2.5-VL-7B-Instruct |
| LLaVA | llava-hf/llava-v1.6-mistral-7b-hf |
| GLM | THUDM/GLM-4.1V-9B-Thinking |

## F.5 EFFECT OF $\lambda_{reg}$

To assess the effect of $\lambda_{reg}$, we applied GHOST to the GLM4.1 Thinking model with $\lambda_{reg} \in \{1.0, 1.5, 2.0\}$. Since the FID score is sensitive to the number of samples, for each pair of $\lambda_{reg}$ values we selected only the samples that were successful in both settings, and then computed the FID score against both the COCO validation set and the original images. As reported in Table 16, increasing $\lambda_{reg}$ consistently makes the generated images more similar to the original ones, leading to improved realism, as reflected by lower FID scores.

## F.6 EFFECT OF $\lambda_{clip}$

Increasing $\lambda_{clip}$ encourages semantic separation between the guiding embedding $z$ and the target-object semantics. We implemented GHOST on two target objects ("traffic light" and "Boat", 200 images each) on Qwen2.5-VL with $\lambda_{clip} \in \{5, 10, 15, 20\}$ and $N{=}4$ samples per image to evaluate the effect of it. As shown in Fig. 14, larger $\lambda_{clip}$ consistently reduces the count of OD-filtered samples, indicating that the regularization term achieves its intended effect by suppressing target-object semantics in the generated images. From $\lambda_{clip} = 5$ to 15, stronger suppression of target-object cues makes it harder for the MLLM to answer "yes", so more samples are generated and also a larger share of generated samples fail to trigger hallucination. At $\lambda_{clip} = 20$, this term becomes over-weighted: fewer runs satisfy $p(y^* \mid \mathbf{X}_q, \Pi(z)) \geq \tau_{yes}$, and more runs hit the maximum (100) steps of optimization (e.g., $10 \rightarrow 20$)

## F.7 EFFECT OF $\tau$

We implemented GHOST with two target objects (200 images each) on LLaVA-1.6, varying $\tau \in \{0.5, 0.6, 0.7, 0.8, 0.9\}$ and using $N{=}4$ generations per image. As shown in Fig. 15, increasing $\tau$ tightens the optimization constraints, and the number of generated samples consistently decreases. At lower $\tau$, the embedding contains weaker target-object cues, so more generations fail to elicit

```
templates = [
    "Do you see a {obj} in the image? Answer with 'Yes' or 'No'.",
    "Is there a {obj} here? Answer with 'Yes' or 'No'.",
    "Does the image contain a {obj}? Answer with 'Yes' or 'No'.",
    "Can you find a {obj} in this picture? Answer with 'Yes' or 'No'.",
    "Would you say there's a {obj} here? Answer with 'Yes' or 'No'.",
    "Is a {obj} present in this image? Answer with 'Yes' or 'No'.",
]
```

Listing 2: Binary question templates used for querying the presence of the target object. The placeholder 'obj' is replaced with the object name (e.g., "boat", "vase") during optimization.

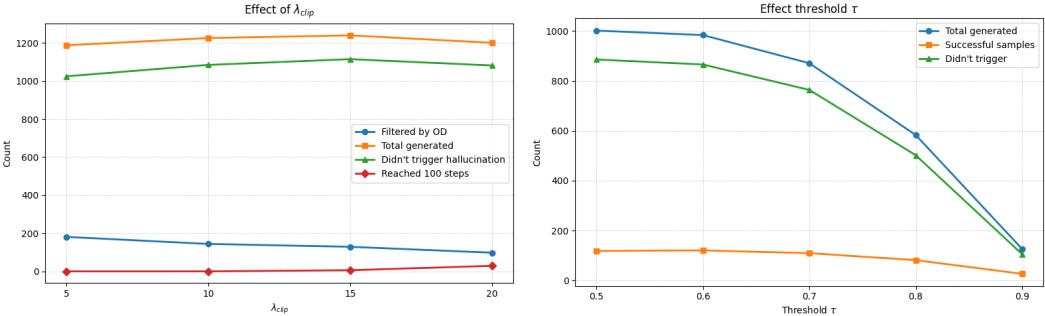

Figure 14: The effect of $\lambda_{\text{clip}}$.      Figure 15: The effect of $\tau$.

Table 16: Ablation study on $\lambda_{\text{reg}}$. FID scores computed on the intersected set of generated images, with respect to the original images and the COCO validation set. The lower FID in each pair is shown in bold.

| Gen. Data | $\lambda_{\text{reg}}$ | Orig | Val |
|-----------|------------|--------|--------|
| (1.0, 1.5) | 1.0 | 127.46 | 145.05 |
|           | 1.5 | **125.11** | **144.11** |
| (1.0, 2.0) | 1.0 | 148.85 | 169.74 |
|           | 2.0 | **138.67** | **163.84** |
| (1.5, 2.0) | 1.5 | 144.25 | 168.20 |
|           | 2.0 | **135.79** | **163.54** |

a "yes" from the MLLM. For $\tau \in \{0.5, 0.6, 0.7\}$, the ratio of total to unsuccessful generations is similar, and the number of successful samples is nearly unchanged.

### F.8  RUNTIME ANALYSIS

We measure the runtime of GHOST across six object categories, using approximately 400 samples per category and running on a single A100 GPU with Qwen2.5-VL as the victim model. Detailed per-class statistics are reported in Table 17. Across all categories, GHOST requires 7–10 seconds per sample, including both embedding optimization and the final image generation. This efficiency stems from optimizing directly in the CLIP embedding space, which results in a short optimization trajectory compared to pixel-space methods. For reference, DASH (Augustin et al., 2025) reports runtimes of roughly 50 seconds per image for PaLI-Gemma-3B and 60 seconds per image for LLaVA-Next-7B, even when using a distilled diffusion model. Thus, GHOST provides a 5–7× speedup per sample on average.

Table 17: Runtime of GHOST across six objects on a single A100 GPU for Qwen2.5-VL.

| Category | Samples | Avg (s) | Min (s) | Max (s) | Total (min) |
|----------|---------|---------|---------|---------|-------------|
| Boat | 378 | 8.59 | 2.23 | 20.04 | 56.20 |
| Bottle | 364 | 9.73 | 2.24 | 19.92 | 61.08 |
| Carrot | 387 | 9.97 | 2.13 | 20.04 | 66.37 |
| Vase | 380 | 10.40 | 2.22 | 20.19 | 67.90 |
| Suitcase | 380 | 9.02 | 2.27 | 20.74 | 59.21 |
| Traffic Light | 398 | 7.18 | 2.24 | 19.60 | 49.66 |

### F.9    STRUCTURAL SIMILARITY INDEX (SSIM)

To further quantify perceptual similarity and assess the preservation of semantic content, we compute the average and standard deviation of SSIM (Wang et al., 2004) across GHOST, unCLIP, and Stable Diffusion outputs using the same setup as our FID semantic fidelity experiment. Higher SSIM indicates greater structural similarity. We obtain:

- **GHOST:** $0.6051 \pm 0.1156$
- **unCLIP:** $0.2322 \pm 0.1218$
- **Stable Diffusion:** $0.1843 \pm 0.1041$

These results indicate that GHOST samples preserve the structural and semantic content of the original images more strongly than standard diffusion outputs.

### F.10    HIGHER RESOLUTION SCALABILITY

To evaluate whether GHOST scales to higher-resolution inputs, we increase the image resolution from 336×336 (the baseline used in the main paper) to 756×756. We train a separate mapper for the higher resolution and run GHOST on two object categories, *traffic light* and *boat*, using 200 samples each on an A100 GPU with Qwen2.5-VL-7B as the victim model. Table 18 reports the runtime statistics and attack success rates.

Across both categories, GHOST continues to run efficiently at the higher resolution. The per-image runtime increases moderately, as expected due to the larger number of visual tokens, while the success rates remain comparable to the 336×336 baseline. These results indicate that GHOST scales to higher resolutions without becoming prohibitively slow. Further speed-ups are possible by parallelizing optimization across samples or adjusting hyperparameters such as the number of optimization steps and $N$.

Table 18: GHOST results on 756×756 resolution input.

| Category | Samples | Avg / Sample (s) | Min (s) | Max (s) | Total (min) | ASR |
|---|---|---|---|---|---|---|
| Traffic Light | 249 | 16.50 | 2.54 | 37.12 | 70.64 | 0.33 |
| Boat | 295 | 9.77 | 2.49 | 37.80 | 50.15 | 0.32 |

## G    GHOST ON OBJECTNET

To evaluate GHOST beyond COCO, we apply our method to 10 categories from ObjectNet (Barbu et al., 2019), using Qwen2.5-VL-7B as the victim model. For each category, we optimize 400 samples, excluding cases where the model already hallucinates the target object (consistent with the main experiments). Table 19 reports the success rates, number of optimized samples, and number of successful samples for all categories. GHOST achieves an overall success of 28%, which is comparable to our results on COCO. Qualitative examples from ObjectNet are shown in Figure 16.

## H    EXTENDING GHOST

GHOST primarily targets object hallucinations. However, to demonstrate its broader potential, we conduct a small experiment to test whether GHOST can also stress-test other types of hallucination, such as attribute or relation hallucinations. Using the same losses as in the main setting, we select samples for which the victim model originally answers "No" to five queries, including "someone is holding a knife," "the vase is transparent," "the bottle is made of glass," "the clock is hanging on the wall," and the relation query "the boat is near the shore." We then optimize the embedding to flip the first-token prediction to "Yes." No object detector is used in this setting, as these queries do not correspond to object presence.

Table 20 summarizes the five queries evaluated, along with the number of selected samples and the final success rates. The average success rate of 32.2% is comparable to the object-existence success

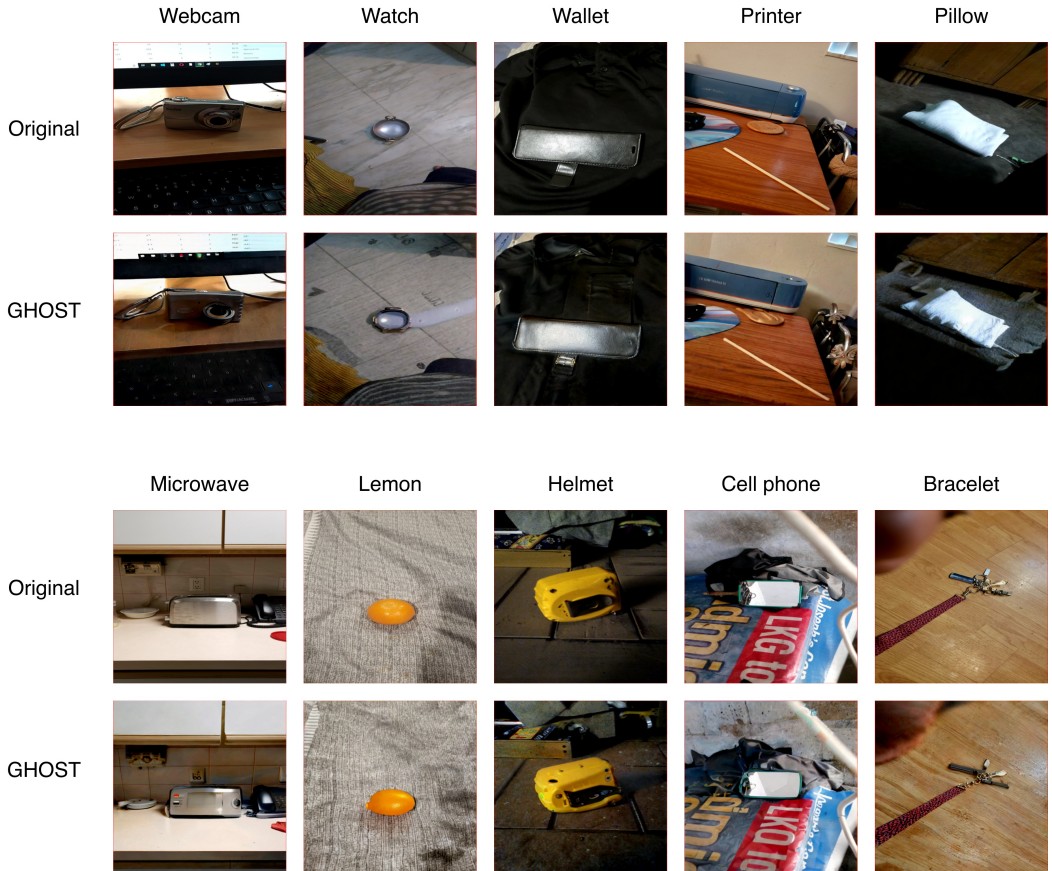

Figure 16: GHOST examples on ObjectNet dataset.

Table 19: GHOST results on 10 ObjectNet categories.

| Category | Samples | Hallucination | Success (%) | Images Generated |
| --- | --- | --- | --- | --- |
| Bracelet | 312 | 100 | 32% | 555 |
| Cellphone | 336 | 134 | 40% | 1017 |
| Helmet | 377 | 124 | 33% | 1259 |
| Lemon | 346 | 80 | 23% | 1205 |
| Microwave | 313 | 38 | 12% | 1128 |
| Pillow | 216 | 102 | 47% | 684 |
| Printer | 357 | 82 | 23% | 1244 |
| Watch | 397 | 67 | 17% | 1244 |
| Wallet | 293 | 141 | 48% | 877 |
| Webcam | 385 | 50 | 13% | 1294 |

rate (29.9%) reported in the main paper. These results provide preliminary evidence that GHOST can extend beyond object-centric hallucinations to induce attribute-level and relation-level failures.

**Hyperparameter Settings**. Table 21 lists the exact optimization hyperparameters used for each query.

**Prompt Templates**. We use two sets of prompt templates: one for attributes and one for relations. These templates are shown in Listing 3.

Table 20: Attribute and relation hallucination experiments. Each query contains only samples for which the model originally answered "No".

| Query Type | Example Question | # Initial Samples | Success |
|---|---|---|---|
| Relation | Is someone holding a knife? | 100 | 16% |
| Relation | Is the clock hanging on the wall? | 74 | 36% |
| Attribute | Is the vase transparent? | 93 | 39% |
| Attribute | Is the bottle made of glass? | 67 | 15% |
| Relation | Is the boat near the shore? | 53 | 55% |
| | | **Average** | **32.2%** |

Table 21: GHOST hyperparameters for attribute and relation hallucination experiments.

| Query | lr | total steps | $\tau$ | N | Guidance Scale | $\lambda_{clip}$ | $\lambda_{reg}$ | OD thr. | t | num_inf. |
|---|---|---|---|---|---|---|---|---|---|---|
| Boat near shore | 0.1 | 100 | 0.8 | 4 | 7.5 | 5.0 | 5.0 | 0.5 | 30 | 50 |
| Someone holding a knife | 0.1 | 100 | 0.8 | 4 | 7.5 | 5.0 | 5.0 | 0.5 | 30 | 50 |
| Bottle made of glass | 0.1 | 100 | 0.8 | 4 | 7.5 | 15.0 | 10.0 | 0.5 | 40 | 50 |
| Clock hanging on wall | 0.1 | 100 | 0.8 | 4 | 7.5 | 5.0 | 5.0 | 0.5 | 30 | 50 |
| Vase is transparent | 0.1 | 100 | 0.8 | 4 | 7.5 | 15.0 | 10.0 | 0.5 | 30 | 50 |

```
# Attribute templates
attribute_templates = [
    "Is the {obj} {attr}? Answer with 'Yes' or 'No'.",
    "Is the {obj} {attr} in the image? Answer with 'Yes' or 'No'.",
    "Would you say the {obj} is {attr}? Answer with 'Yes' or 'No'.",
    "Can you confirm the {obj} is {attr}? Answer with 'Yes' or 'No'.",
    "Does the {obj} appear {attr}? Answer with 'Yes' or 'No'.",]

# Relation templates
relation_templates = [
    "{rel}? Answer with 'Yes' or 'No'.",
    "Is it true that {rel}? Answer with 'Yes' or 'No'.",
    "Can you confirm that {rel}? Answer with 'Yes' or 'No'.",
    "Does the image show that {rel}? Answer with 'Yes' or 'No'.",
    "Would you say that {rel}? Answer with 'Yes' or 'No'.",]
```

Listing 3: Prompt templates used for attribute-based and relation-based hallucination experiments. The placeholders 'obj', 'attr', and 'rel' are filled with the corresponding object name, attribute phrase, or relational statement during optimization.

## I   OBJECT DETECTOR USAGE

In GHOST, the object detector serves only as a sanity-check filter to remove images that may still contain the target object. The detector does not participate in the optimization process, all losses operate entirely in the CLIP embedding space and are designed to remove object-specific information. To ensure that detector false negatives do not impact our results, we additionally perform a human evaluation confirming that the retained samples do not contain the target object (Sec. 5.1). Since the optimization itself is detector-agnostic, GHOST does not depend on any particular detector family.

**Ensemble experiment.** To further assess detector dependence, we apply a strict ensemble filter using four detectors spanning both open-vocabulary and closed-set families: OWLv2 (Minderer et al., 2023), GroundingDINO (Liu et al., 2023b), YOLO-World (Cheng et al., 2024), and YOLOv8 (Jocher et al., 2023). A generated sample is discarded if any detector assigns confidence $\geq 0.5$ to the target object. We evaluate 300 samples across five object classes using Qwen2.5-VL as the victim model. The results are reported in Table 22: even under this conservative filtering, GHOST maintains a 22.7% success across 1,419 images.

Table 22: Ensemble of Object Detectors: A sample is removed if any of four detectors predict the target object with confidence $\geq 0.5$.

| Object | Success | Samples | Hallucination |
|--------|---------|---------|---------------|
| Boat | 16% | 279 | 45 |
| Bottle | 25% | 271 | 68 |
| Clock | 15% | 290 | 44 |
| Suitcase | 29% | 280 | 81 |
| Traffic Light | 28% | 299 | 84 |
| **Overall** | **22.7%** | **1419** | **322** |

## J  FINETUNING

**Setup**. We fine-tune Qwen2.5-VL-7B on its GHOST-generated images to assess whether such counterfactual samples improve model performance. This is intended as a proof-of-concept demonstration, with large-scale training and full-class coverage left for future work. We use LoRA (Hu et al., 2022) for parameter-efficient adaptation, BitsAndBytes (Dettmers et al., 2023) for quantization, and TRL (von Werra et al., 2020) for SFT implementation. All experiments are conducted on an 8-way L40S GPU node. Full configuration details are provided in Table 23.

**Dataset**. We construct a balanced fine-tuning set by sampling 150 negative images (GHOST images) and generating 150 positive images for each class, resulting in 3,000 samples in total. Positive samples are synthesized using Stable Diffusion unCLIP (Rombach et al., 2022), conditioned on images that contain the target object. We avoid using real COCO images as positives so that the model cannot rely on a trivial shortcut such as "synthetic = negative, real = positive." We use the prompt: *Is there a {obj} in the image? Answer with 'Yes' or 'No'.* with "Yes" for positives and "No" for negatives.

To increase label fidelity for the synthesized positives, we rely on COCO annotations and filter candidates by ranking their CLIP embeddings against a compositional text embedding of the target object (Sec. F.2), keeping the highest-scoring images as inputs to the unCLIP model. This improves the likelihood that the positive set truly depicts the intended object while keeping the data domain consistent (synthetic vs. synthetic). Example positive samples are shown in Fig. 17.

**Model Selection**. Since LoRA fine-tuning can be sensitive to hyperparameters, we train multiple configurations and select the best-performing model on POPE (that is reported in Table 23). We generally observe that the rank, learning rate, and batch size are the most influential factors for effective fine-tuning.

**Model Evaluation**. We evaluate hallucination robustness using POPE under its *Random* evaluation setting. We also evaluate caption-level hallucination using CHAIR (Rohrbach et al., 2018), which measures the proportion of objects mentioned in the generated caption but absent from the image. Following Li et al. (2023), we adopt both $CHAIR_i$ and $CHAIR_s$, which quantify hallucination at the object-instance and sentence level respectively. We generate captions using the prompt *"Generate a short caption of the image."* and evaluate on 1,000 samples from the COCO validation split. As shown in Table 4, GHOST fine-tuning yields consistent improvements on both POPE and CHAIR.

To assess the impact of fine-tuning on overall model ability, we evaluate the model on both VQA and image captioning. For VQA, we use the small VQAv2 subset from Hugging Face[2], which contains image–question pairs with a single-word ground-truth answer. We report accuracy for both the base and fine-tuned models. For captioning, we use the COCO validation images and their annotations as ground truth. We generate captions using the prompt *"Write a short caption for the given image."* for both models and compute BERTScore (Zhang* et al., 2020) between the generated captions and COCO references. These evaluations measure whether GHOST fine-tuning impacts general capabilities beyond hallucination robustness.

---

[2]https://huggingface.co/datasets/merve/vqav2-small

Table 23: Fine-tuning configuration for Qwen2.5-VL.

| Quantization (BitsAndBytes) | |
| --- | --- |
| 4-bit Quantization | NF4 |
| Compute Dtype | `bfloat16` |
| Quant Storage | `bfloat16` |
| Double Quant | True |
| **LoRA Configuration** | |
| Rank ($r$) | 2 |
| Alpha | 2 |
| Dropout | 0.05 |
| Target Modules | all-linear |
| Modules Saved | lm_head, embed_tokens |
| **Training Setup (SFTConfig)** | |
| Optimizer | AdamW (fused) |
| Learning Rate | $1 \times 10^{-5}$ |
| Epochs | 6 |
| Batch Size | 8 |
| Grad. Accumulation | 1 |
| Precision | `bf16` |
| Seed | 42 |

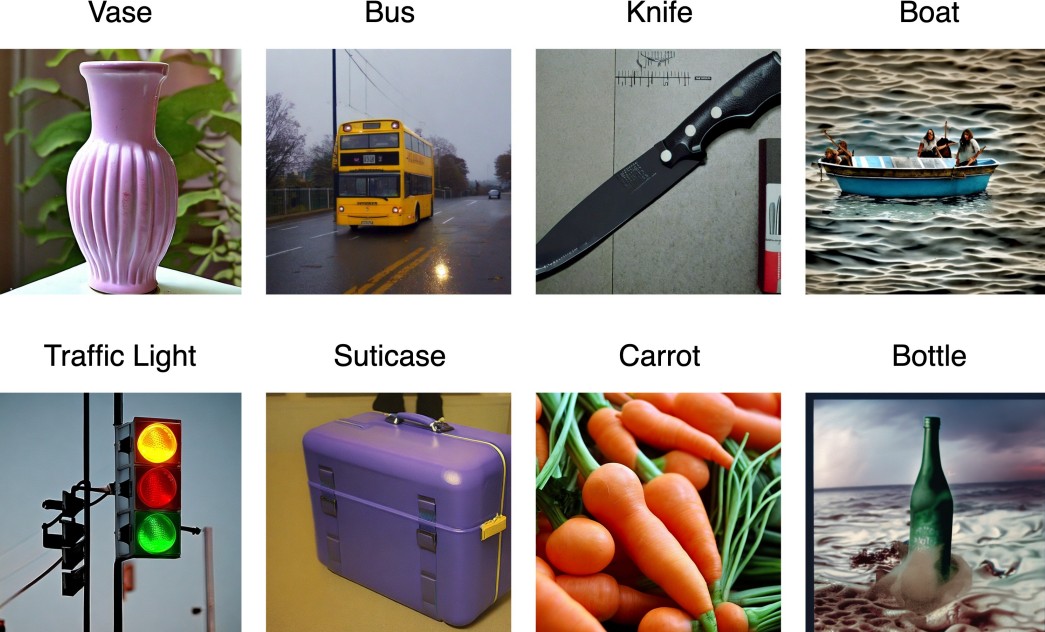

Figure 17: Positive samples generated with Stable Diffusion unCLIP (Rombach et al., 2022)

## K  GHOST ADDITIONAL RESULTS

In this section, we provide extended qualitative and quantitative results to complement the main paper. Figures 18, 19, and 20 present qualitative examples of GHOST applied to Qwen2.5, LLava-1.6, and GLM-4.1Thinking, respectively, showing the original images alongside GHOST-generated samples and the corresponding model responses. Tables 24, 25, and 26 report detailed sample statistics for each object category across the three victim models, including the number of generated, filtered, and successful samples. Finally, Tables 27, 28, and 29 provide class-wise transferability results, highlighting how GHOST-induced hallucinations on one victim model transfer to others.

Is there a boat in this image? Answer with yes or no, then explain shortly.

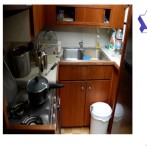 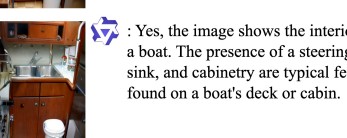

: No, there is no boat in the image. The image shows a kitchen area with a stove, sink, and some kitchen utensils. There are no boats visible in this scene.

: Yes, the image shows the interior of a boat. The presence of a steering wheel, sink, and cabinetry are typical features found on a boat's deck or cabin.

Is there a clock in this image? Answer with yes or no, then explain shortly.

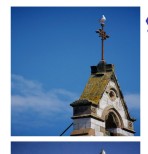

:No, there is no clock in the image. The structure appears to be a small tower or spire with a cross at the top and some architectural details, but there is no visible clock face or mechanism.

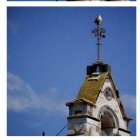

:Yes, there is a clock in the image. It is located on the right side of the structure and appears to be part of a decorative architectural element.

Is there a bus in this image? Answer with yes or no, then explain shortly.

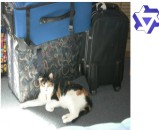

: No, there is no bus in the image. The image shows a cat lying on the floor next to some suitcases and a patterned bag. There are no vehicles, including buses, visible in the picture.

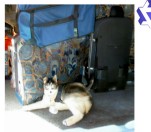

: Yes, there is a bus in the image. The interior of the vehicle shows seats and a patterned wall that is characteristic of a bus, along with a cat lying on the floor, which suggests it might be a pet-friendly bus or a converted bus into a living space.

Is there a knife in this image? Answer with yes or no, then explain shortly.

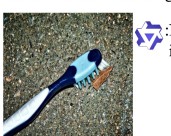

:No, there is no knife in the image. The object present is a toothbrush with a piece of paper stuck to it.

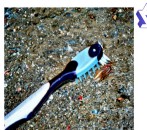

:Yes, there is a knife in the image. The object has a handle and a blade, which are characteristic features of a knife. The blade appears to be modified or decorated with an eye-like design, but it still retains the essential shape and structure of a knife.

Is there a traffic light in this image? Answer with yes or no, then explain shortly.

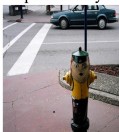

: No, there is no traffic light in the image. The image shows a fire hydrant painted to look like a person wearing a hat and a yellow jacket, with a car and a street in the background.

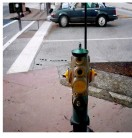

: Yes, there is a traffic light in the image. It appears to be a pedestrian crossing signal, which is used to control pedestrian traffic at intersections. The signal has a yellow housing and is mounted on a pole, indicating it's designed for pedestrians to cross safely when the light is green.

Is there a suitcase in this image? Answer with yes or no, then explain shortly.

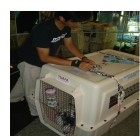

:No, there is not a suitcase in the image. The object in question appears to be a pet carrier, which is used for transporting animals, as indicated by the presence of what looks like a small animal inside and the design of the container.

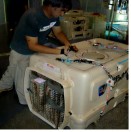

:Yes, there is a suitcase in the image. It appears to be a large, white, hard-shell suitcase with various items and decorations attached to it, including what looks like a small cage or container on top of it.

Is there a carrot in this image? Answer with yes or no, then explain shortly.

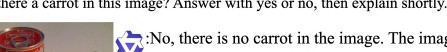

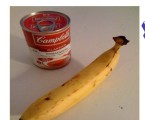

:No, there is no carrot in the image. The image shows a can of Campbell's tomato soup and a banana.

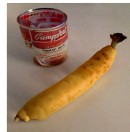

:Yes, there is a carrot in the image. The object on the right side of the image has the characteristic shape and color of a carrot, including its root end and leafy top.

Is there a toilet in this image? Answer with yes or no, then explain shortly.

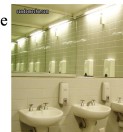

:No, there is no toilet in the image. The image shows a row of sinks with soap dispensers and mirrors above them, which are typical features in a public restroom or bathroom setting.

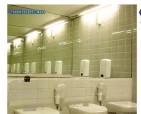

:Yes, there is a toilet in the image. The image shows a public restroom with multiple urinals and a toilet visible on the left side of the frame.

Figure 18: Qualitative samples from GHOST on Qwen2.5 as the victim model, along with Qwen's responses to each image. For each pair, the top image shows the original input, while the bottom image shows the corresponding sample generated by GHOST.

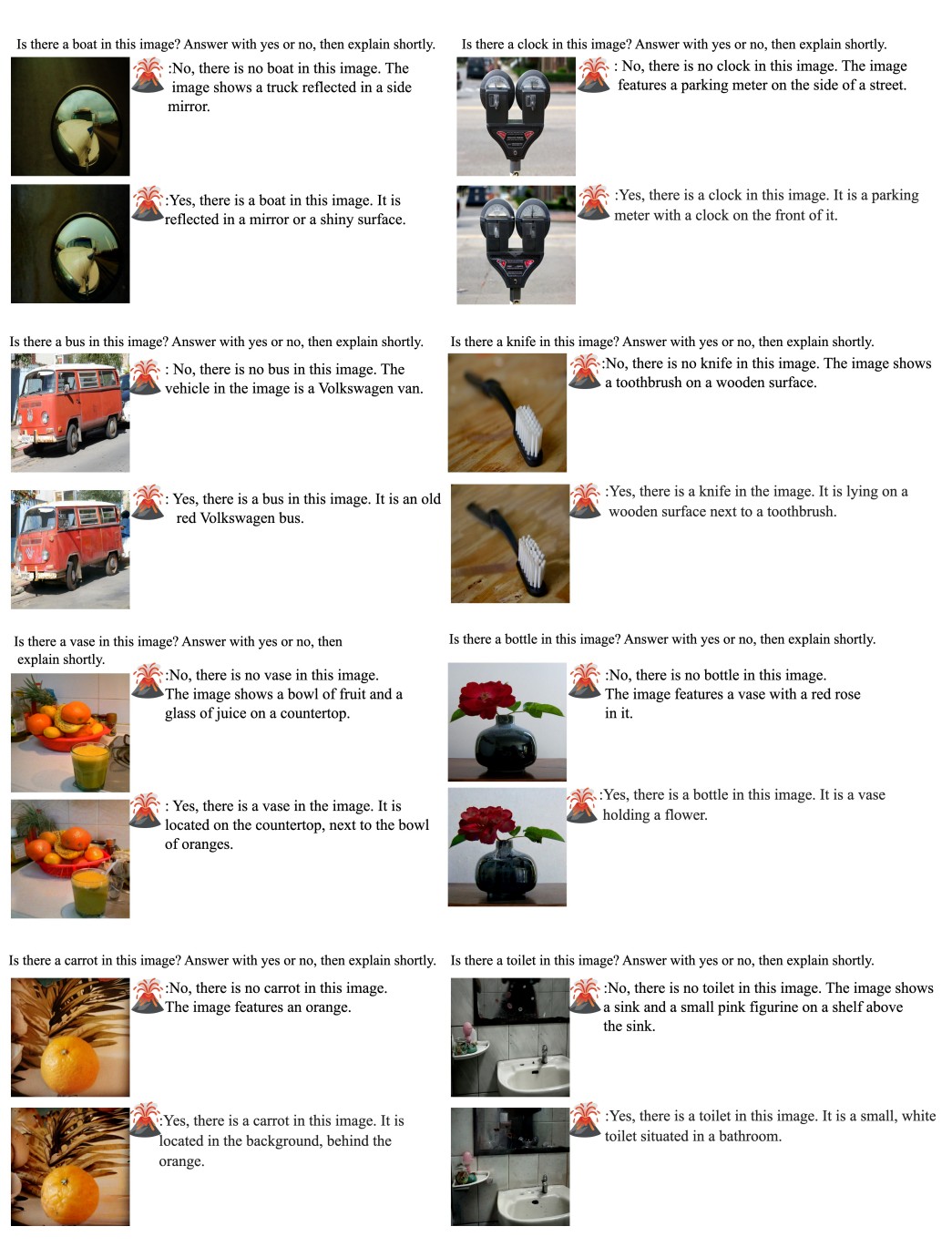

Figure 19: Qualitative samples from GHOST on LLava1.6 as the victim model, along with LLava's responses to each image. For each pair, the top image shows the original input, while the bottom image shows the corresponding sample generated by GHOST.

Target object: Vase

:Got it, let's look at the image. The object shown is an elephant-shaped figurine, not a vase. Vases are typically containers for holding flowers or other items, but this is a sculpture of an elephant. So the answer should be No.

:Got it, let's look at the image. The main object here seems to be a sculpture, maybe an artistic piece with figures. Wait, is there a vase? Wait, the sculpture might be a vase-shaped object? Wait, no, let me check again. Wait, the image shows a sculpture that might be a vase, but let's confirm. Wait, the question is if there's a vase. Let me check the details. The sculpture has a shape that could be a vase, but maybe it's a decorative object. Wait, the image has a sculpture that might be a vase. Wait, no, maybe not. Wait. ... so the answer is Yes.

Target object: Boat

:Got it, let's look at the image. The scene shows a bridge with a "STOP" sign, ropes, and a dock area. There's no boat visible in the image. So the answer should be No.

:Got it, let's look at the image. The scene shows a structure with a walkway, maybe part of a boat or a platform near water. The setting seems to be on a boat or a floating structure. So, is there a boat? Let's check. The area with the walkway and the water suggests it's part of a boat. So the answer would be Yes

Target object: Toilet

Got it, let's look at the image. The question is asking if there's a toilet. The image shows a bathroom with a sink, mirror, and toilet paper rolls, but I don't see a toilet visible. Wait, maybe the toilet is not in the frame? Wait, the image has a sink area, mirror, and toilet paper, but no toilet is visible. So the answer would be No.

:Got it, let's look at the image. There's a white toilet visible near the sink area. So the answer should be Yes.

Target object: Knife

:Got it, let's look at the image. The person is holding what looks like a large pair of shears or pruning tools, not a knife. Wait, the question is about a knife. Let me check again. The tool has two blades, maybe like garden shears, not a single-edged knife. So there's no knife visible. Therefore, the answer is No

:Got it, let's look at the image. The person is holding a large tool, which looks like a knife or a sword. Wait, the object has a blade, so it's a knife-like weapon. So the answer should be Yes

Target object: Carrot

:Got it, let's look at the picture. The question is asking if there's a carrot. Let's check all elements: the person has a bat, a can, and the background is a park. There's no carrot visible. So the answer is No.

: Got it, let's look at the image. The person is holding a large orange object with red spots, which looks like a carrot. Wait, is it a carrot? The object has the shape and color of a carrot, so I need to check. The question is whether there's a carrot. The object in the left hand is orange with red, maybe a carrot. So the answer would be Yes.

Target object: Suitcase

:Got it, let's look at the image. The container is a white fabric box, maybe a storage bin, not a suitcase. Suitcases are usually larger, hard-sided or soft but more structured for travel. This is a small storage box under a piece of furniture. So the answer should be No.

:Got it, let's look at the image. There's a beige bag with text, maybe a suitcase? Wait, the question is about a suitcase. The object behind the teddy bear looks like a suitcase or a bag. So is there a suitcase? Let's check. The image has a bag with "Glui" text, maybe a suitcase. So the answer would be Yes.

Figure 20: Qualitative samples from GHOST on GLM4.1Thinking as the victim model, along with GLM's thinking phase in each image. For each pair, the top image shows the original input, while the bottom image shows the corresponding sample generated by GHOST.

Table 24: Sample statistics for different objects with Qwen2.5 as the victim model.

| Object | Samples Considered | Image Generated | Image Filtered | Successful Samples |
|--------|-------------------|-----------------|----------------|-------------------|
| Boat | 915 | 3141 | 191 | 258 |
| Bottle | 881 | 2746 | 21 | 383 |
| Bus | 891 | 2995 | 311 | 244 |
| Carrot | 956 | 3353 | 132 | 228 |
| Clock | 962 | 3473 | 392 | 175 |
| Knife | 966 | 2368 | 114 | 353 |
| Suitcase | 949 | 3067 | 234 | 367 |
| Toilet | 969 | 3368 | 817 | 289 |
| Traffic light | 992 | 3291 | 424 | 312 |
| Vase | 942 | 3279 | 101 | 207 |

Table 25: Sample statistics for different objects with LLlava-1.6 as the victim model.

| Object | Samples Considered | Image Generated | Image Filtered | Successful Samples |
|--------|-------------------|-----------------|----------------|-------------------|
| Boat | 863 | 2804 | 63 | 270 |
| Bottle | 763 | 1558 | 4 | 221 |
| Bus | 786 | 2221 | 168 | 178 |
| Carrot | 948 | 3041 | 101 | 218 |
| Clock | 927 | 2588 | 198 | 298 |
| Knife | 843 | 1092 | 29 | 188 |
| Suitcase | 885 | 1644 | 100 | 364 |
| Toilet | 945 | 3081 | 543 | 268 |
| Traffic light | 967 | 2508 | 160 | 241 |
| Vase | 859 | 1499 | 33 | 216 |

Table 26: Sample statistics for different objects with GLM-4.1 Thinking as the victim model.

| Object | Samples Considered | Image Generated | Image Filtered | Successful Samples |
|--------|-------------------|-----------------|----------------|-------------------|
| Boat | 847 | 2032 | 202 | 305 |
| Bottle | 710 | 1346 | 53 | 348 |
| Bus | 764 | 2102 | 373 | 321 |
| Carrot | 939 | 3130 | 76 | 272 |
| Clock | 937 | 3146 | 297 | 290 |
| Knife | 968 | 2644 | 80 | 271 |
| Suitcase | 937 | 2322 | 276 | 328 |
| Toilet | 942 | 1932 | 486 | 217 |
| Traffic light | 979 | 2294 | 192 | 176 |
| Vase | 866 | 2137 | 141 | 286 |

Table 27: Class-wise transferability results with Qwen2.5 as the victim model.

| Object | LLaMA 3.2 | LLava-1.6 | GLM-4.1 |
|--------|-----------|-----------|---------|
| Boat | 67.8 | 68.2 | 79.7 |
| Bottle | 67.1 | 66.1 | 85.4 |
| Bus | 73.0 | 49.6 | 85.4 |
| Carrot | 52.2 | 48.3 | 64.7 |
| Clock | 57.7 | 70.3 | 75.6 |
| Knife | 64.0 | 64.9 | 52.7 |
| Suitcase | 69.2 | 70.3 | 58.7 |
| Toilet | 73.4 | 52.9 | 73.5 |
| Traffic light | 67.7 | 64.4 | 78.1 |
| Vase | 58.9 | 61.8 | 71.7 |

Table 28: Class-wise transferability results with Llava-1.6 as the victim model.

| Object | LLaMA 3.2 | Qwen2.5 | GLM-4.1 |
|---|---|---|---|
| Boat | 51.9 | 56.7 | 51.9 |
| Bottle | 50.2 | 55.2 | 63.2 |
| Bus | 48.9 | 47.2 | 49.0 |
| Carrot | 47.3 | 57.3 | 57.6 |
| Clock | 41.3 | 52.0 | 53.3 |
| Knife | 52.1 | 52.1 | 37.4 |
| Suitcase | 50.6 | 47.0 | 32.0 |
| Toilet | 60.1 | 55.2 | 54.7 |
| Traffic light | 47.7 | 51.0 | 57.1 |
| Vase | 46.8 | 53.7 | 56.5 |

Table 29: Class-wise transferability results with GLM-4.1Thinking as the victim model.

| Object | LLaMA3.2 | Qwen2.5 | LLava1.6 |
|---|---|---|---|
| Boat | 75.7 | 67.0 | 66.0 |
| Bottle | 75.6 | 72.5 | 62.8 |
| Bus | 62.6 | 42.7 | 35.8 |
| Carrot | 45.2 | 52.2 | 31.6 |
| Clock | 56.1 | 56.1 | 65.9 |
| Knife | 84.5 | 69.0 | 68.6 |
| Suitcase | 82.2 | 70.9 | 68.1 |
| Toilet | 69.4 | 68.9 | 48.4 |
| Traffic light | 71.7 | 72.8 | 53.9 |
| Vase | 64.6 | 63.2 | 64.9 |

