# OpenReview forum: "GHOST: Hallucination-Inducing Image Generation for Multimodal LLMs"
_ICLR.cc/2026/Conference — ICLR 2026 Poster_

### Official Review · Reviewer_GNtm · 2025-10-27

**Soundness:** 3
**Presentation:** 3
**Contribution:** 3
**Rating:** 6
**Confidence:** 3

**Summary:**

This paper introduces GHOST, a method for automatically generating images designed to induce object hallucinations in VLM. Unlike static benchmarks , GHOST actively synthesizes images by optimizing a CLIP image embedding to elicit a hallucinated response while simultaneously regularizing against inserting the object's semantics and staying close to an original image embedding . The authors demonstrate GHOST's effectiveness in inducing hallucinations across various MLLMs, including reasoning models, and show that these hallucinations are highly transferable, even to closed-source models like GPT-4o. Finally, preliminary experiments suggest that finetuning an MLLM on GHOST-generated images can improve its robustness against hallucinations on GHOST's own generated data and the POPE benchmark.

**Strengths:**

1.The work constructs a benchmark for VLM hallucination vulnerability from a novel perspective by optimizing visual embeddings, demonstrating effective hallucination-inducing attacks against sota open-source models. Furthermore, it shows that hallucinations induced on open-source models can effectively transfer to sota closed-source VLMs, suggesting shared vulnerabilities.

2.The paper demonstrates that finetuning models using the data generated by GHOST can reduce object hallucination, showing improvements on both the GHOST-generated adversarial images and established hallucination benchmarks like POPE.

**Weaknesses:**

1.A significant concern is the potential distribution gap between the images decoded from the optimized visual embeddings and natural images. For example, the right image in Figure 1 appears somewhat unnatural . It is questionable whether using such potentially out-of-distribution images constitutes a fair or meaningful way to attack and evaluate the robustness of MLLMs against real-world hallucinations.

2.The experiments validating the mitigation effect of finetuning on GHOST data seem insufficient. While results on POPE and GHOST's own data are shown, the evaluation would be more convincing if it included results on other recognized hallucination benchmarks such as CHAIR or HallusionBench to demonstrate broader robustness improvements.

**Questions:**

1.GHOST's attack generation involves an adversarial loss tailored to a specific victim model . Given the observed transferability, could a potentially universal hallucination dataset be constructed by optimizing against a set of diverse open-source models? Could such a dataset be effectively used for evaluating and perhaps even finetuning other models to improve general hallucination robustness?

2.The approach of finetuning on GHOST images, adversarially generated images causing hallucination, conceptually resembles methods like PerturboLLaVA, which finetunes on perturbed text to mitigate hallucinations. Could the mathematical framework or explanation provided by PerturboLLaVA be adapted to theoretically explain why finetuning on GHOST's visually perturbed, hallucination-inducing images improves the model's resistance to object hallucination?

---

> ### Author Response · Authors · 2025-11-21
> **Response to Reviewer GNtm**
>
> We thank the reviewer for recognizing GHOST’s novel embedding-based attack, its strong transferability across models, and the benefit of our finetuning results. Below, we address the reviewer’s points:
>
> ## Weakness 1
> We thank the reviewer for raising this concern. We believe GHOST-generated images remain sufficiently in-distribution and realistic for the diagnostic and corrective applications proposed in this work. As reported in Table 2 (Semantic Fidelity), the FID scores indicate that GHOST samples preserve the distribution of COCO images, and the “Distributional Realism” metrics show that they maintain general perceptual quality.
>
> To further support this, we conducted an additional experiment measuring SSIM between the original images and the outputs of GHOST, unCLIP, and SDv2 (using Qwen2.5-VL-7B as the victim model). The results are:
>
> **GHOST:** 0.6051 ± 0.1156
> **unCLIP:** 0.2322 ± 0.1218
> **Stable Diffusion:** 0.1843 ± 0.1041
>
> These scores show that GHOST preserves the structural and semantic content of the initial samples far more strongly than standard diffusion outputs. This aligns with GHOST’s design goal: introducing hallucination cues while keeping the overall scene semantics intact.
>
> Importantly, the finetuning results demonstrate that these images are meaningful for robustness training. Finetuning on GHOST samples improves performance on real-image hallucination benchmarks such as POPE and CHAIR, confirming that the generated images are not off-distribution artifacts but effective for learning more robust behavior.
>
> ---
>
> ## Weakness 2
> To address reviewer’s concern, we re-ran and updated our fine-tuning results by adding CHAIR, following the setup of Li et al. [1], and observe consistent improvements after GHOST fine-tuning. We will include all evaluation details in the revised version.
>
> |        | POPE | CHAIRs | CHAIRi | captions | vqa2  |
> |--------|------|--------|--------|----------|-------|
> | BaseQwen | 88.7 | 3.8    | 3      | **89.5**     | **72.83** |
> | +ft      | **93.2** | **2.9**    | **2.6**    | 89.4     | 71.51 |
>
> [1] Li et al., Evaluating Object Hallucination in Large Vision-Language Models.
>
> ## Question 1
> We thank the reviewer for raising this interesting direction. While we could not construct and evaluate such a universal dataset within the rebuttal timeframe, our results suggest that this is indeed feasible. From our original and added results in this rebuttal, it is highly likely that the universal hallucination dataset can be successfully constructed, and it is likely to improve the general hallucination robustness. The main limitation is computational, building a universal attack requires obtaining feedback from multiple large MLLMs during each optimization step, which increases memory and compute requirements. Nevertheless, we agree that this is a promising and plan to explore it in future work.
>
> ## Question 2
> **Theoretical explanation (connection to PerturboLLaVA).**
> Yes, the theoretical framework of PerturboLLaVA extends naturally to our setting. Their decomposition of “true signal vs. bias signal” can be applied in *image space* by splitting the input into core-evidence pixels and spurious-context pixels.
> We denote:
>
> $I_c: \text{core evidence}, \qquad
> I_s: \text{spurious cues (background, textures, co-occurrence patterns)}.$
>
> Thus,
>
> $p(x_1 \mid I) = p(x_1 \mid I_c, I_s).$
>
> Following the same Bayes-style factorization used in PerturboLLaVA, we obtain:
>
> $p(x_1 \mid I_c, I_s)
> \,\propto\,
> p(x_1 \mid I_c)\,\frac{p(x_1 \mid I_s)}{p(x_1)}.$
>
> This matches PerturboLLaVA’s decomposition into “true-evidence” and “bias” terms.
>
> ---
>
> **How GHOST fits this framework.**
> GHOST constructs adversarial images where the object is absent \(I_c = 0\) but the spurious cues \(I_s\) are automatically amplified by the optimization. In practice, GHOST introduces visual patterns that the model over-relies on, such as sharpened edges resembling a knife or hour–hand-like structures for clocks.This increases:
>
> $p_\theta(x_1=\text{``yes''} \mid I_s^{\text{adv}}),$
>
> which directly boosts the spurious-cue term in the factorization above.
>
> **Why finetuning on GHOST improves robustness.**
> During finetuning, we apply the correct supervision:
>
> $L_{\text{GHOST}}=-\log p_\theta(x_1=\text{``no''} \mid I_{\text{adv}}).$
>
> Because the model cannot directly modify the prior over spurious cues   $p(x_1\mid I_s)$, the only way to satisfy the supervision is to reduce reliance on spurious signals and increase reliance on core evidence.
>
> This is the same mechanism described in PerturboLLaVA, except that GHOST operates on *visual* spurious features rather than textual ones.

---

> > ### Comment · Reviewer_GNtm · 2025-11-26
> >
> > Thank you for the clarification and analysis. Most of my concerns have been addressed, and I will maintain my score.

---

### Official Review · Reviewer_8Ved · 2025-10-29

**Soundness:** 3
**Presentation:** 3
**Contribution:** 3
**Rating:** 4
**Confidence:** 4

**Summary:**

The paper proposes GHOST (Generating Hallucinations via Optimizing Stealth Tokens), a pipeline for stress‑testing multimodal large language models (MLLMs) by automatically generating images that induce object hallucinations. Starting from an input image and a target object absent from that image, GHOST optimizes the image’s CLIP embedding to increase the MLLM’s probability of answering “yes” to a query about the target’s presence, while regularizing the embedding to remain close to the original and discouraging direct encoding of the object’s semantics. To avoid backpropagating through the diffusion model and the MLLM, the authors train a mapper (a simple MLP) that aligns the CLIP vision embedding space with the MLLM’s vision encoder, enabling efficient gradient updates. Once the optimized embedding reaches a confidence threshold, an unCLIP diffusion model generates an image conditioned on this embedding, starting from a partially noised latent of the original image to preserve overall structure. An open‑vocabulary detector filters out images where the target object actually appears, ensuring that only hallucinations are counted.

**Strengths:**

S1. GHOST introduces a principled pipeline to generate hallucination‑inducing images.

S2. Images optimized for one model induce hallucinations in others, highlighting shared failure modes across architectures.

S3. Fine‑tuning on GHOST images reduces hallucinations and improves POPE scores while retaining performance on VQAv2 and captioning.

**Weaknesses:**

W1. Experiments focus on 10 COCO object classes. The method may perform differently on a broader set of objects, particularly those with abstract or ambiguous semantics. The human evaluation sample is limited to 40 volunteers, which may not capture diverse perceptions.

W2. The OWLv2 detector filters out samples where the target object is present, but open‑vocabulary detectors are not perfect; subtle insertions might slip through, biasing success rates.

W3. Fine‑tuning is demonstrated only on Qwen2.5‑VL with synthetic positives and negatives. More robust experiments (e.g., across models or with real images) and analysis of downstream safety implications would strengthen the claim that GHOST can act as a corrective tool.

W4. quantify the per‑image runtime or GPU requirements; scaling to many objects or higher resolutions could be prohibitive.

W5. GHOST hinges on the availability of CLIP vision embeddings and an unCLIP diffusion model. It is unclear whether other vision encoders or diffusion architectures would offer similar alignment.

**Questions:**

Q1. What is the average time to generate a hallucination‑inducing image, and how does this scale with image resolution and the number of target objects? Could the process be parallelized or approximated to reduce cost?

Q2. Have you tested GHOST with alternative diffusion architectures (e.g., SDXL, DiT) or other vision encoders (e.g., EVA CLIP)? Would the mapper still align the embedding spaces effectively?

Q3. How sensitive are the results to the choice of OWLv2? Could false negatives in detection inflate success rates? Did you manually inspect filtered samples to ensure the target object was truly absent?

---

> ### Author Response · Authors · 2025-11-21
> **Response to Reviewer 8Ved**
>
> We thank the reviewer for recognizing GHOST’s principled pipeline, its strong cross-model transferability, and the effectiveness of our finetuning results. Below, we address the reviewer’s points:
>
> ## Weakness 1
> Number of selected classes: Our goal in selecting the original 10 COCO classes was to cover diverse object types. To further test generality beyond COCO, we additionally conducted experiments on 10 new categories from ObjectNet[5]. GHOST continues to achieve strong success rates in this domain:
>
> | Object    | ASR  | Optimized | Images Generated | Successful |
> |-----------|------|-----------|------------------|------------|
> | Bracelet  | 0.32 | 312       | 555              | 100        |
> | Cellphone | 0.40 | 336       | 1017             | 134        |
> | Helmet    | 0.33 | 377       | 1259             | 124        |
> | Lemon     | 0.23 | 346       | 1205             | 80         |
> | Microwave | 0.12 | 313       | 1128             | 38         |
> | Pillow    | 0.47 | 216       | 684              | 102        |
> | Printer   | 0.23 | 357       | 1244             | 82         |
> | Watch     | 0.17 | 397       | 1244             | 67         |
> | Wallet    | 0.48 | 293       | 877              | 141        |
> | Webcam    | 0.13 | 385       | 1294             | 50         |
>
> These results show that GHOST generalizes well to new objects and a different visual distribution, not just the COCO subset used in the main paper.
>
> ### Regarding Human Evaluation Sample Size
> We strongly believe that our human evaluation follows standard practice. The number of participants (40) is well within the range used in prior multimodal hallucination and image-evaluation studies. For example, works such as [1–4] report human-evaluation studies involving 10–30 participants, and in several cases even fewer.
>
> ---
>
> ## Weakness 2, Question 3
> To ensure that our success rates are not inflated by detector false negatives, we performed a human-evaluation study in which volunteers manually inspected GHOST-generated images. This confirmed that the generated samples do not contain the target object, verifying that our filtering process is not overlooking subtle insertions. Details about the human evaluation is in the paper section 5.1.
>
> We also conducted another experiment to further ensure that our method does not rely on a single detector, we repeated the filtering using an ensemble of four detectors spanning both open-vocabulary and closed-set families: OWLv2 (open-vocab), GroundingDINO (open-vocab), YOLO-World (open-vocab), YOLOv8 (closed-set). A sample was removed if any detector predicted the object with ≥ 0.5 confidence.
>
> We evaluated 300 samples across 5 classes, yielding:
>
> | Object        | ASR  | Total Images | Successful |
> |---------------|------|--------------|------------|
> | Boat          | 0.16 | 279          | 45         |
> | Bottle        | 0.25 | 271          | 68         |
> | Clock         | 0.15 | 290          | 44         |
> | Suitcase      | 0.29 | 280          | 81         |
> | Traffic Light | 0.28 | 299          | 84         |
>
> ### Overall ensemble-filtered performance
> | Metric                   | Value |
> |--------------------------|-------|
> | Total successful samples | 322   |
> | Total images             | 1419  |
> | Global ASR               | 22.7% |
>
> Even under this strict multi-detector filtering, GHOST still achieves a 22.7% success rate, demonstrating that our results are not due to detector failures and that the attack remains highly effective without relying on any single detector.
>
> ---
>
> ## Weakness 3
> Please note that our fine-tuning is on synthetic images, while our hallucination evaluation benchmarks (POPE and CHAIR) use real images, and we observe improvements. This demonstrates that the generated images are sufficiently natural and informative for practical robustness enhancement. This aligns with prior works [7,8,9], which show that diffusion-generated samples can effectively improve classifier robustness on real-world images.
>
> To address the reviewer's concern, we re-ran and updated our fine-tuning results by adding the CHAIR hallucination benchmark, following the setup and prompts from Li et al. [6]. GHOST fine-tuning yields consistent improvements on CHAIR as well. We will include all evaluation and implementation details in the revised version.
>
> |        | POPE | CHAIRs | CHAIRi | captions | vqa2  |
> |--------|------|--------|--------|----------|-------|
> | BaseQwen | 88.7 | 3.8    | 3      | **89.5**     | **72.83** |
> | +ft      | **93.2** | **2.9**    | **2.6**    | 89.4     | 71.51 |

---

> ### Author Response · Authors · 2025-11-21
> **Response to Reviewer 8Ved**
>
> ## Weakness 4 Question1
> ### Runtime analysis
> We conducted a detailed runtime analysis across 6 object categories using the Qwen2.5-VL-7B model.
>
> | Category      | Total Runtime (s) | Total Runtime (min) | Samples | Avg / Sample (s) | Min (s) | Max (s) |
> |---------------|-------------------|----------------------|---------|-------------------|---------|---------|
> | Boat          | 3372.20           | 56.20               | 378     | 8.59              | 2.23    | 20.04   |
> | Bottle        | 3664.77           | 61.08               | 364     | 9.73              | 2.24    | 19.92   |
> | Carrot        | 3982.47           | 66.37               | 387     | 9.97              | 2.13    | 20.04   |
> | Vase          | 4073.92           | 67.90               | 380     | 10.40             | 2.22    | 20.19   |
> | Suitcase      | 3552.89           | 59.21               | 380     | 9.02              | 2.27    | 20.74   |
> | Traffic Light | 2979.33           | 49.66               | 398     | 7.18              | 2.24    | 19.60   |
>
> Across all categories, GHOST consistently requires 7–10 seconds per sample, including both embedding optimization and final generation process. We believe GHOST is efficient and scalable due to bridging the embedding space directly, enabling a much shorter optimization trajectory. By contrast, DASH reports:
> - ~50 seconds per image for PaLI-Gemma 3B
> - ~60 seconds per image for LLaVA-Next 7B
> while using a distilled diffusion model.
>
> Thus, GHOST is approximately 5–7× faster per sample. We will include these in the paper as well.
>
> ---
>
> ### Higher-resolution scalability
> We believe GHOST is scalable to higher-resolution images. To demonstrate this, we increased the resolution from **336×336** (the baseline in the paper) to **756×756**, trained a separate mapper for the higher resolution, and re-ran the method. We tested this on two objects, **traffic light** and **boat**, using 200 samples each on an A100 GPU and Qwen model as victim.
>
> The results are shown below:
>
> | Object        | Total Runtime (s) | Total Runtime (min) | Samples Processed | Avg / Sample (s) | Min (s) | Max (s) | Attack Success Rate |
> |---------------|-------------------|----------------------|--------------------|-------------------|---------|---------|----------------------|
> | Traffic Light | 4238.42           | 70.64               | 249                | 16.50            | 2.54    | 37.12   | 0.33                 |
> | Boat          | 3008.99           | 50.15               | 295                | 9.77             | 2.49    | 37.80   | 0.32                 |
>
> This result shows that GHOST continues to run efficiently at significantly higher resolutions. The per-image runtime increases moderately, consistent with the expected computational cost of higher number of token embeddings, and the attack success rate remains comparable to the 336×336 baseline. Thus, GHOST is scalable to higher-resolution images and the method does not become prohibitively slow at larger resolutions.
>
> Further speed-ups are also possible by parallelizing the optimization across images and adjusting the hyperparameters (e.g., N and the total number of GHOST steps) when needed.
>
> ## Weakness 5 Question 2
> The embedding-space mapping itself is not tied to any specific vision encoder. In our experiments, we successfully optimized embeddings for:
> **Qwen2.5-VL** (Qwen2.5 vision transformer),
> **LLaVA-1.6** (CLIP ViT-L/14),
> **GLM-v4.1** (AIMv2 backbone).
>
> These models use different vision encoders with different embedding, yet our mapper consistently aligned them with the unCLIP diffusion embedding space. This provides empirical evidence that the alignment method generalizes across architectures. Our choice of unCLIP is primarily driven by the design choice of GHOST: the diffusion model must support direct conditioning on a vision embedding so that optimization can be performed entirely in embedding space. To the best of our knowledge, this capability is currently not available in SDXL, DiT, or most other diffusion variants, making unCLIP the most promising practical choice for our setting. However, we expect it to extend to other architectures as long as the diffusion model allows embedding-level conditioning.

---

> ### Author Response · Authors · 2025-11-21
> **Response to Reviewer 8Ved**
>
> 1. Taihang Hu, Linxuan Li, Joost van de Weijer, Hongcheng Gao, Fahad Shahbaz Khan, Jian Yang, Ming-Ming Cheng, Kai Wang, Yaxing Wang, Token Merging for Training-Free Semantic Binding in Text-to-Image Synthesis, NeurIPS2024
> 2. Xiefan Guo, Jinlin Liu, Miaomiao Cui, Jiankai Li, Hongyu Yang, Di Huang, InitNO: Boosting Text-to-Image Diffusion Models via Initial Noise Optimization, CVPR2024
> 3. Tuna Han Salih Meral, Enis Simsar, Federico Tombari, Pinar Yanardag, CONFORM: Contrast is All You Need For High-Fidelity Text-to-Image Diffusion Models, CVPR2024
> 4. Zhennan Chen, Yajie Li, Haofan Wang, Zhibo Chen, Zhengkai Jiang, Jun Li, Qian Wang, Jian Yang, Ying Tai, Region-Aware Text-to-Image Generation via Hard Binding and Soft Refinement, ICCV2025
> 5. Andrei Barbu, David Mayo, Julian Alverio, William Luo, Christopher Wang, Dan Gutfreund, Josh Tenenbaum, and Boris Katz. Objectnet: A large-scale bias-controlled dataset for pushing the limits of object recognition models. In Advances in Neural Information Processing Systems 32
> 6. Li et al., Evaluating Object Hallucination in Large Vision-Language Models.
> 7. Jianhao Yuan, Francesco Pinto, Adam Davies, and Philip Tor, Not Just Pretty Pictures: Toward Interventional Data Augmentation Using Text-to-Imager, ICML2024
> 8. Maan Qraitem,, Kate Saenko , and Bryan A. Plummer, From Fake to Real: Pretraining on Balanced Synthetic Images to Prevent Spurious Correlations in Image Recognition, ECCV2024
> 9. Aryan Yazdan Parast, Basim Azam, Naveed Akhtar, DDB: Diffusion Driven Balancing to Address Spurious Correlations, ICCV2025

---

> ### Comment · Reviewer_8Ved · 2025-11-22
>
> Thank you for the detailed rebuttal and for the substantial set of additional experiments. These additions meaningfully strengthen several aspects of the work. That said, several concerns remain.
>
> Although the new ObjectNet experiments and the attribute/relation hallucination results expand the evaluation, the overall coverage of object types and hallucination forms remains fairly narrow relative to the strong general claims about “stress-testing” MLLMs. The diagnostic aspect (targeted generation of hallucination-inducing inputs) is well demonstrated, but the broader claim of providing a comprehensive hallucination robustness benchmark feels less fully substantiated. The attack’s generality beyond object-centric settings would benefit from more systematic treatment.
>
> The corrective angle is intriguing, and the added CHAIR results strengthen it. However, fine-tuning is only demonstrated on a single architecture (Qwen2.5-VL-7B) and primarily on object-existence hallucinations. While the preliminary evidence is promising, the scope is limited, and the framing should more clearly reflect this. As it stands, the corrective claim feels somewhat stronger than what the evidence fully supports.
>
> The method’s reliance on diffusion models that support embedding-level conditioning (such as unCLIP) is a practical limitation. While the authors note that this is not a conceptual constraint, in practice it restricts the applicability of GHOST to a relatively small class of models. This dependency should be acknowledged explicitly as a limitation rather than implied to generalize automatically to architectures lacking this capability.
>
> Overall, given the remaining questions, the submission still feels borderline relative to ICLR’s standard. With more modest framing of the broader claims and careful integration of the new evidence, I believe the work has the potential for meaningful impact.

---

> ### Author Response · Authors · 2025-11-24
>
> > Thank you for the detailed rebuttal and for the substantial set of additional experiments. These additions meaningfully strengthen several aspects of the work. That said, several concerns remain.
>
> **Response:** We appreciate the acknowledgment of the additional experiments and the strengthened aspects of the work. We address the remaining concerns below.
>
> ---
>
> > Although the new ObjectNet experiments and the attribute/relation hallucination results expand the evaluation, the overall coverage of object types and hallucination forms remains fairly narrow relative to the strong general claims about “stress-testing” MLLMs. The diagnostic aspect (targeted generation of hallucination-inducing inputs) is well demonstrated, but the broader claim of providing a comprehensive hallucination robustness benchmark feels less fully substantiated. The attack’s generality beyond object-centric settings would benefit from more systematic treatment.
>
> **Response:** Thank you for this thoughtful observation. We agree that the current scope of hallucination types in the paper is limited, and we will revise the framing accordingly. Our work is primarily focused on **object-existence hallucination**, and we will adjust the manuscript to make this scope explicit and more modest. The additional attribute and relation experiments included in the rebuttal phase were intended to illustrate the *potential* capabilities and extensibility of GHOST, but not to claim comprehensive coverage. In the revised submission, we will clearly state that the core contribution of the paper lies in diagnosing and inducing object-existence hallucinations, and we will position broader hallucination types (attributes, relations, and beyond) as promising avenues for future work.
>
> ---
>
> > The corrective angle is intriguing, and the added CHAIR results strengthen it. However, fine-tuning is only demonstrated on a single architecture (Qwen2.5-VL-7B) and primarily on object-existence hallucinations. While the preliminary evidence is promising, the scope is limited, and the framing should more clearly reflect this. As it stands, the corrective claim feels somewhat stronger than what the evidence fully supports.
>
> **Response:** We appreciate this precise assessment. As noted in the paper, the corrective setup we present is a *small toy setup* designed to demonstrate the potential of GHOST for mitigating hallucinations, rather than a full corrective framework. We will revise the framing to emphasize that these results represent *initial evidence*, and we will highlight broader corrective evaluations as future work.
>
> ---
>
> > The method’s reliance on diffusion models that support embedding-level conditioning (such as unCLIP) is a practical limitation. While the authors note that this is not a conceptual constraint, in practice it restricts the applicability of GHOST to a relatively small class of models. This dependency should be acknowledged explicitly as a limitation rather than implied to generalize automatically to architectures lacking this capability.
>
> **Response:** Thank you for this valuable recommendation. We will make this dependency explicit in the paper.
>
> To address the reviewer’s concerns, we will add the following **Limitations and Future Work** section to the paper:
>
> *A key practical limitation of GHOST is its reliance on diffusion models that support image embedding-level conditioning, such as unCLIP. GHOST optimizes directly in the vision-embedding space, which requires a generative model capable of decoding modified embeddings back into pixel space. Many popular diffusion architectures do not currently provide this interface and therefore cannot serve as drop-in replacements in our framework. In addition, our current evaluation primarily focuses on object-centric hallucinations; a more systematic investigation across a broader range of hallucination modalities, along with more extensive corrective evaluations across architectures and hallucination types, would provide a more comprehensive assessment.*
>
>
> ---
>
> > Overall, given the remaining questions, the submission still feels borderline relative to ICLR’s standard. With more modest framing of the broader claims and careful integration of the new evidence, I believe the work has the potential for meaningful impact.
>
> **Response:** We appreciate the constructive feedback. As outlined above, we will revise the manuscript to adopt more modest and precise framing, and we will integrate the new evidence carefully. We are grateful for the acknowledgment that the work has meaningful potential.

---

> > ### Comment · Reviewer_8Ved · 2025-11-27
> >
> > Thank you for the response. While the empirical scope and corrective evidence remain limited, I appreciate the revised framing and added experiments. I will slightly increase my score.

---

### Official Review · Reviewer_PFEa · 2025-11-01

**Soundness:** 2
**Presentation:** 2
**Contribution:** 2
**Rating:** 4
**Confidence:** 2

**Summary:**

This paper introduces a new method for inducing object hallucination in MLLMs.
To address this issue, the paper utilizes an image generation model to automatically create images that preserve the semantic meaning of the original inputs while still being able to trigger hallucinations.
Through extensive experiments, the method demonstrates a significantly higher hallucination success rate compared to previous approaches across multiple models, confirming its effectiveness.
Additionally, the paper shows that images generated for one model can induce hallucinations in other models, and that fine-tuning with generated data helps mitigate hallucination, highlighting its dual role as both a diagnostic and corrective tool.

**Strengths:**

* Unlike prior benchmarks relying on static datasets,  this approach dynamically generates images that preserve the original semantics while triggering hallucinations, providing a fully automated way.

* The framework is well-designed and clearly presented. Each component is justified, and the ablation studies effectively demonstrate their contributions.

* The experiments are extensive and evaluate on a diverse set of MLLMs using various metrics.

**Weaknesses:**

Major Weakness

* **Insufficient Motivation for Human-Perceptible Modifications:** While GHOST introduces “human-perceptible” changes, the paper does not explain why human perceptibility is a necessary design choice. This aspect distinguishes GHOST from adversarial methods but is presented without motivation.

* **Unclear Justification for Using SD unCLIP as the Semantic Preservation Backbone:** The paper adopts SD unCLIP as the image generation backbone to preserve semantic content from the original image, yet it does not clearly justify why this model was chosen over other diffusion variants.
  Moreover, Table 2 shows inconsistent FID scores across models: While SD unCLIP achieves low FID values for some settings, in other cases its FID is high. This raises questions about whether SD unCLIP is a reliable method for maintaining semantic similarity.

* **Fairness of Comparison with DASH:** Table 1 compares GHOST with DASH, yet the evaluation settings differ significantly. Although both methods share the same goal of inducing hallucinations via false-positive responses, DASH is fundamentally designed for large-scale, open-world datasets (e.g., ReLAION-5B) to detect *systematic hallucination patterns* across diverse object classes. In contrast, GHOST operates on a COCO-based subset. Given this difference in scale and design philosophy, directly comparing hallucination success rates on COCO may not be fair.

Minor Weakness

* **Clarity in Figure 3**
  * The black dashed arrow is ambiguous; please clarify what operation or information flow it represents.

* **Typos and Formatting Issues**
  * Line 235–236: In Equation (2), the two terms involving X\_v​ appear inconsistent.

  * Appendix H.2: There seems to be an empty placeholder for *unCLIP*.

**Questions:**

* Could you clarify why *human-perceptible* modifications were chosen as a design goal? What advantages do these perceptible cues provide over imperceptible perturbations in evaluating hallucination?

* Why was **SD unCLIP** selected as the semantic preservation backbone instead of other diffusion variants? Table 2 shows inconsistent FID results across models, suggesting that SD unCLIP may not always guarantee semantic fidelity

* In Table 1, the results may not be entirely fair since GHOST and DASH were designed with different objectives. How could the two methods be compared in a more balanced and fair manner?

---

> ### Author Response · Authors · 2025-11-21
> **Response to Reviewer PFEa**
>
> We thank the reviewer for their positive assessment of our work, the clear design and ablations, and the breadth of our experimental evaluation. At the same time, we believe some concerns arise from misunderstandings of our core motivations and contributions, which we clarify below.
>
> ## Weakness 1 / Question 1:
> ### Human-Perceptible Modification
> Our goal is to uncover model-specific, semantically meaningful visual failure modes, rather than to jailbreak the model or find pixel-level adversarial perturbations. GHOST is designed to produce subtle semantic, human-perceptible misleading cues (such as vase-like textures or object–context co-occurrences) that reveal what visual cue the model relies on. These cues expose shortcut behavior grounded in spurious correlations, which is essential for diagnosing hallucinations. In contrast, adversarial perturbations inject imperceptible pixel-level noise that flips predictions without revealing interpretable or semantic failure modes.
>
> This design choice also supports mitigation, not just diagnosis. As shown in Section 6, fine-tuning on GHOST-generated images improves performance on real hallucination benchmarks, demonstrating that models learn to overcome genuine semantic shortcuts. Fine-tuning on adversarial examples, by contrast, improves robustness only to adversarial noise and does not help on natural images.
>
> Finally, while both adversarial examples and GHOST-generated samples exhibit transferability across models, the underlying reasons differ. Adversarial transfer arises from similarities in high-dimensional decision boundaries, whereas GHOST transfer stems from shared visual shortcut reliance across MLLMs.
>
>
>
> ## Weakness 2 / Question 2:
> ### Choice of SD unCLIP
> Our optimization is performed directly in the image embedding space, rather than pixel space, because the embedding space provides a rich semantic representation while allowing efficient gradient-based updates. This design requires a diffusion backbone that can be additionally guided by image embedding signals, not only by text prompts. Among open-source models, SD-unCLIP is uniquely suited for this. Our use of SD-unCLIP is therefore driven by architectural compatibility with GHOST’s optimization pipeline, not by a claim that SD-unCLIP universally offers the best FID performance. Any model with SD-unCLIP's ability of added image embedding guidance can potentially replace SD-unCLIP in our technique.
>
> ### Regarding FID variability
> The FID variations in Table 2 do not indicate instability of SD-unCLIP. As explained in the paper, we report two distinct FID metrics:
>
> 1. **Distributional Realism**, which compares generated images to the validation distribution
> 2. **Semantic Fidelity**, which compares generated images to the original images to assess semantic preservation
>
> Because SD-unCLIP is conditioned directly on the original image embedding, it naturally obtains lower (better) Semantic Fidelity FID than Distributional Realism FID, consistent with its strong semantic preservation (Fig. 6). Importantly, the same pattern holds for SD-v2: SD-v2 also has lower Semantic Fidelity FID than Distributional Realism FID, since for SD-v2 we compute Semantic Fidelity by comparing images generated from MS-COCO captions to their corresponding original images. Thus, the relative ordering of these two metrics reflects how the FID is defined, not instability specific to SD-unCLIP. Moreover, for Distributional Realism, SD-unCLIP and SD-v2 achieve very similar FID scores, indicating that SD-unCLIP is not less realistic or less stable than SD-v2.
>
> To ensure fairness given FID’s sensitivity to sample size, we compute each model’s FID on the subset of images for which that model’s attack succeeds. Because the successful-attack sets differ across models, the FID values are necessarily computed on different sample subsets. Therefore, differences in FID across models reflect differences in the successful-attack samples, not inconsistency or instability of SD-unCLIP. Importantly, within each model, the trends we report remain consistent with what we claim in the paper.

---

> > ### Author Response · Authors · 2025-11-21
> > **Response to Reviewer PFEa**
> >
> > ## Weakness 3 / Question 3:
> > ### Comparison with DASH
> > We respectfully note that our intention regarding the DASH comparison is already stated in Lines 348–353. Both methods share the high-level goal of identifying model-specific hallucination failures, but they operate in different regimes. DASH is designed to search for naturally occurring false positives across extremely large real-image corpora. Its original implementation requires scanning ReLAION-5B, and even on that massive dataset the DASH paper reports very low success rates; the same low rates appear on COCO in our experiments.
> >
> > In contrast, GHOST does not rely on naturally occurring hallucinations. It directly constructs targeted, semantically consistent adversarial images that reliably induce hallucinations, allowing us to obtain many controlled failure cases without requiring billion-scale data. Comparison with DASH highlights that relying solely on naturally occurring hallucination samples is insufficient, and GHOST addresses this gap by generating high-quality, targeted examples.
> >
> > ---
> >
> > ## Minor weaknesses
> >
> > ### Figure 3
> > The black dashed arrow in Figure 3c represents the forward and backward passes of the diffusion process. Specifically, we first obtain the latent embedding of the input image to get z_0, add noise for several steps to obtain z_t, and then apply the standard denoising process to generate the final image. We will clarify this more in the revised version.
> >
> > ### Appendix H.2
> > We thank the reviewer for pointing out this typo, and we will correct it in the revised version.

---

### Official Review · Reviewer_6FPP · 2025-11-01

**Soundness:** 3
**Presentation:** 2
**Contribution:** 2
**Rating:** 2
**Confidence:** 4

**Summary:**

The paper introduces GHOST, a gradient-free method that generates images designed to induce hallucinations in text-to-image and multimodal models, and establishes a benchmark for measuring hallucination robustness. Experiments show that GHOST achieves high attack success rates and strong transferability across multiple open-source models, revealing both model-specific weaknesses and shared vulnerabilities. The authors also explore simple mitigation strategies, such as fine-tuning, which help reduce but not fully eliminate the hallucination effect, demonstrating GHOST’s potential as a tool for evaluating and improving model robustness.

**Strengths:**

* The paper proposes an effective model-specific attack that generates adversarial images capable of inducing hallucinations in a target MLLM regarding a given target object in the original image.

* The generated images demonstrate strong transferability, degrading the performance of other models beyond the specific target model. This suggests that the proposed approach reveals not only model-specific weaknesses but also shared vulnerabilities across models.

**Weaknesses:**

* The proposed method is limited to inducing misperception of object existence and does not address other types of hallucination, such as attribute or relationship hallucinations. This limitation is reflected in Table 4, where improvements are observed mainly on POPE—a benchmark focused on object-centric VQA—while no significant gains are found on VQAv2 or Caption benchmarks. This implies that the method may not generalize to mitigating broader hallucination types.

* The adversarially generated images (e.g., Figures 1, 2, 4, and 6) appear somewhat unnatural. This raises concerns about the practical impact of improving robustness against such images in real-world applications that primarily involve natural images. From an impact perspective, it would strengthen the work if the human evaluation included not only whether the target object was preserved or removed, but also whether the generated images looked natural to human observers. The current reliance on FID scores is insufficient to fully assess perceptual naturalness.

* The proposed method requires multiple optimization steps per image, which may reduce its efficiency. A quantitative analysis of this efficiency—such as reporting the hit ratio (i.e., success rate among optimized samples)—would be helpful.

* The approach involves numerous hyperparameters (e.g., thresholds and control factors), which implies that substantial model-specific tuning may be required to achieve optimal attack performance. This limits its scalability and reproducibility.

* Minor issue: The paper could improve notation consistency—for example, the fonts for λ_clip and λ_reg differ in several equations.

**Questions:**

* The method regularizes the generated image to remain close to the original image (Equation 3). From an attack perspective, it is unclear why maintaining similarity to the original image is important, since the primary goal is to induce hallucination in the model. A clearer justification for this design choice would be valuable.

* It would be interesting to see how the proposed method performs on image domains other than COCO, as this would test the generalization and robustness of GHOST across more diverse visual distributions.

---

> ### Author Response · Authors · 2025-11-21
> **Response to Reviewer 6FPP**
>
> We thank the reviewer for acknowledging the effectiveness of our model-specific attack, for noting the strong transferability of the generated images across models, and for recognizing GHOST’s value as a tool for revealing shared vulnerabilities. Below, we address the points raised by the reviewer:
> ## Weakness 1
> ### **Grounding Objects**
> We believe accurate object grounding is a prerequisite for more complex multimodal reasoning and downstream tasks. We therefore center GHOST on object hallucination. This setting is well studied, with standard benchmarks such as POPE and CHAIR, and prior work including DASH, Che et al. [1], Sarkar et al. [2], and Leng et al. [3] directly targeting it. However, GHOST is not inherently limited to object-existence hallucinations, and our technique is generalizable.
>
> ### **GHOST is Generalizable**
> We conducted additional experiments showing that GHOST generalizes to attribute and relation hallucinations as well.
> In these settings, we pick samples for which the victim model originally answers “No,” then apply GHOST to induce a hallucinated “Yes.”
>
> Below we summarize four such experiments:
>
> | Question (Attribute / Relation)        | Success | Initial samples selected |
> |----------------------------------------|---------|---------------------------|
> | Is someone holding a knife?            | 16%     | 100                       |
> | Is the clock hanging on the wall?      | 36%     | 74                        |
> | Is the vase transparent?               | 39%     | 93                        |
> | Is the bottle made of glass?           | 15%     | 67                        |
>
> The average ASR of 26.5 is comparable to the ASR 29.9 achieved in the paper for object hallucination. These results demonstrate that GHOST is capable of producing attribute-level and relation-level hallucinations beyond simple object existence. We will include the details of the experiment with qualitative samples to the main paper.
>
> ### Improvements on Benchmark
> Our fine-tuning results show clear gains on POPE and CHAIR (we added this benchmark to address reviewer GNtm), which directly measure object-level hallucinations, the grounding failure that GHOST is designed to expose. In contrast, VQAv2 and Caption are not hallucination benchmarks; we include them only to verify that general capabilities remain stable, following DASH. Because GHOST is diagnostic in nature and not primarily a fine-tuning method, we expect improvements to concentrate on hallucination benchmarks rather than broad VQA or captioning tasks.
>
> ## Weakness 2
> ### **Naturalness of GHOST Images**
> Our view is that the quality of GHOST-generated samples is sufficient for both diagnostic and corrective applications, which are the primary purposes of this work. The goal is not to create perfectly photorealistic images, but rather to generate semantically meaningful perturbations that expose model vulnerabilities. The FID results in the paper serve to show that GHOST preserves image quality of the underlying diffusion model.We chose FID because it is a widely used metric for this purpose. Nevertheless, to address the reviewer's concern, we now further validate the naturalness of our images with human evaluation.
>
> ### **Human Evaluation for Naturalness**
> We conducted a human evaluation study comparing 30 random GHOST images with unCLIP-generated images guided by the same initial samples. Volunteers rated images on the following 1–5 naturalness scale:
>
> 1 — Completely unnatural
> 2 — Somewhat unnatural
> 3 — Neutral
> 4 — Mostly natural
> 5 — Completely natural
>
> The results are:
>
> | Model        | Average Rating | #Votes | # ≥ 4 (“mostly natural”) |
> |--------------|----------------|--------|---------------------------|
> | GHOST        | 3.55           | 191    | 111                       |
> | SD-unCLIP    | 3.56           | 189    | 118                       |
>
> These ratings show that GHOST images are perceived as comparably natural to diffusion-model baseline. These results will be mentioned in the revised paper.
>
> ### **SSIM** (Structural Similarity Index)
> Moreover, to address any of the reviewer's remaining concerns about insufficiency of quantitative evaluation of perceptual quality, We also performed a complementary experiment using average and std of SSIM (Structural Similarity Index) to measure preservation of semantic content and proximity to COCO distribution. Using the same setup as our FID semantic fidelity on GHOST, unCLIP and SD images, we obtained (higher means more similarity):
>
> **GHOST:** 0.6051 ± 0.1156
> **unCLIP:** 0.2322 ± 0.1218
> **Stable Diffusion:** 0.1843 ± 0.1041
>
> These results indicate that GHOST samples preserve the semantic structure of the original image more strongly than standard diffusion outputs.

---

> ### Author Response · Authors · 2025-11-21
> **Response to Reviewer 6FPP**
>
> ### **Qualitative results**
> Qualitative results in the paper actually demonstrate sufficiently natural images. We do want to emphasize that perceptual 'unnaturalness' is being exaggerated in the paper because we also provide reference to natural images right beside the perturbed images. We request the reviewer to also consider naturalness without using the original images as reference. We also believe qualitative results in the main text and appendix further demonstrate that the images remain sufficiently natural for humans to intuitively understand why the model hallucinates (e.g., sharpened “knife-like’’ edges, addition of an hour hand to induce clock hallucination).
>
> Finally, prior works [5, 6, 7] show that diffusion-generated samples, can be effectively leveraged to improve classifier robustness on real-world images. Consistent with this, our fine-tuning results on POPE and CHAIR, which evaluate hallucinations on real images, show improvements. This confirms that the generated images are sufficiently natural and informative for practical robustness enhancement.
> ## Weakness 3
> ### **Runtime analysis**
> We conducted a detailed runtime analysis across 6 object categories using the Qwen2.5-VL-7B model.
>
> | Category      | Total Runtime (s) | Total Runtime (min) | Samples | Avg / Sample (s) | Min (s) | Max (s) |
> |---------------|-------------------|----------------------|---------|-------------------|---------|---------|
> | Boat          | 3372.20           | 56.20               | 378     | 8.59              | 2.23    | 20.04   |
> | Bottle        | 3664.77           | 61.08               | 364     | 9.73              | 2.24    | 19.92   |
> | Carrot        | 3982.47           | 66.37               | 387     | 9.97              | 2.13    | 20.04   |
> | Vase          | 4073.92           | 67.90               | 380     | 10.40             | 2.22    | 20.19   |
> | Suitcase      | 3552.89           | 59.21               | 380     | 9.02              | 2.27    | 20.74   |
> | Traffic Light | 2979.33           | 49.66               | 398     | 7.18              | 2.24    | 19.60   |
>
> Across all categories, GHOST consistently requires 7–10 seconds per sample, including both embedding optimization and final generation process. We believe GHOST is efficient due to bridging the embedding space directly, enabling a much shorter optimization trajectory. By contrast, DASH reports:
> - ~50 seconds per image for PaLI-Gemma 3B
> - ~60 seconds per image for LLaVA-Next 7B
> while using a distilled diffusion model.
>
> Thus, GHOST is approximately 5–7× faster per sample. We will include these in the paper as well.
>
> ### **Hit ratio**
> Regarding the request for a **hit ratio** the full statistics, including the total number of generated samples and the number of successful attacks for all three models—are already reported in Appendix Tables 16, 17, and 18. These tables provide complete breakdowns of optimized samples, filtered samples, and successful hallucinations.
>
> ## Weakness 4
> ### **Reproducibility**
> To ensure reproducibility, we provide complete and explicit hyperparameter settings for all components of GHOST in the Appendix (Appendix F.3 for optimization parameters, Appendix D.1 for fine-tuning, and Appendix H.1 for the mapper MLP). In addition, we will publicly release the full source code, making all experimental configurations directly accessible. While optimal hyperparameters can further improve performance, our ablation studies show that GHOST is not brittle to hyperparameter choices. Across a wide range of settings, the method continues to achieve strong success rates. Hence, scalability is not an issue.
>
> ## Minor Issue
> Thank you for pointing these out. We will address them in the final version.
>
> ## Question 1
> GHOST is not designed as an unconstrained attack. We aim to inject hallucination cues while preserving the overall semantics of the original image, allowing GHOST to stress-test models and serve as a diagnostic tool. The regularization term helps maintain that semantic structure. Also, an unbounded perturbation to the original image can lead to distorted images, which may compromise the meaningfulness of any diagnostics our approach is used for.
>
> As shown in the FID ablation in Appendix Table 15, $\lambda_{reg}$ (the regularization term) keeps the generated sample closer to the input image, improving naturalness and realism, and keeping the outputs in-distribution with COCO images.

---

> > ### Author Response · Authors · 2025-11-21
> > **Response to Reviewer 6FPP**
> >
> > ## Question 2
> > To evaluate GHOST beyond COCO, we applied our method to 10 categories from ObjectNet[8], using Qwen2.5-VL-7B as the target model. For each category, we optimized 400 samples (samples which the model is already hallucinating are excluded, the same as main experiments). The results are shown below:
> >
> > | Object    | ASR  | Optimized Samples | Images Generated | Successful Samples |
> > |-----------|------|-------------------|------------------|--------------------|
> > | Bracelet  | 0.32 | 312               | 555              | 100                |
> > | Cellphone | 0.40 | 336               | 1017             | 134                |
> > | Helmet    | 0.33 | 377               | 1259             | 124                |
> > | Lemon     | 0.23 | 346               | 1205             | 80                 |
> > | Microwave | 0.12 | 313               | 1128             | 38                 |
> > | Pillow    | 0.47 | 216               | 684              | 102                |
> > | Printer   | 0.23 | 357               | 1244             | 82                 |
> > | Watch     | 0.17 | 397               | 1244             | 67                 |
> > | Wallet    | 0.48 | 293               | 877              | 141                |
> > | Webcam    | 0.13 | 385               | 1294             | 50                 |
> >
> > These results confirm that GHOST generalizes well to other visual distributions, and retains strong attack performance on ObjectNet. The overall ASR is 0.28, which is comparable to our results on COCO and demonstrates the generalizability of the attack. These results along with qualitative samples from the dataset will be included in the revised version.
> >
> > [1]Liwei Che and Tony Qingze Liu and Jing Jia and Weiyi Qin and Ruixiang Tang and Vladimir Pavlovic, Hallucinatory Image Tokens: A Training-free EAZY Approach on Detecting and Mitigating Object Hallucinations in LVLMs, ICCV2025
> > [2] Pritam Sarkar, Sayna Ebrahimi, Ali Etemad, Ahmad Beirami, Sercan Arik, Tomas Pfister, Mitigating Object Hallucination in MLLMs via Data-augmented Phrase-level Alignment, ICLR2025
> > [3] Sicong Leng, Hang Zhang, Guanzheng Chen, Xin Li, Shijian Lu, Chunyan Miao, Lidong Bing, VCD: Mitigating Object Hallucinations in Large Vision-Language Models through Visual Contrastive Decoding, CVPR2024
> > [5] Jianhao Yuan, Francesco Pinto, Adam Davies, and Philip Tor, Not Just Pretty Pictures: Toward Interventional Data Augmentation Using Text-to-Imager, ICML2024
> > [6] Maan Qraitem,, Kate Saenko , and Bryan A. Plummer, From Fake to Real: Pretraining on Balanced Synthetic Images to Prevent Spurious Correlations in Image Recognition, ECCV2024
> > [7] Aryan Yazdan Parast, Basim Azam, Naveed Akhtar, DDB: Diffusion Driven Balancing to Address Spurious Correlations, ICCV2025
> > [8] Andrei Barbu, David Mayo, Julian Alverio, William Luo, Christopher Wang, Dan Gutfreund, Josh Tenenbaum, and Boris Katz. Objectnet: A large-scale bias-controlled dataset for pushing the limits of object recognition models. In Advances in Neural Information Processing Systems 32

---

> > > ### Comment · Reviewer_6FPP · 2025-11-24
> > > **Thank you for your response**
> > >
> > > Thank you for addressing my comments and questions. In particular, the additional analysis on naturalness and runtime helps alleviate some of my concerns.
> > >
> > > That said, I still have remaining reservations. My primary concern is that the proposed method appears to remain largely limited to object hallucination, and I am not fully convinced of the practicality of stress-testing models using artificially perturbed images.
> > >
> > > Although you argue—based on a few examples—that GOHST is not restricted to object hallucination, the provided examples seem highly dependent on single objects and are quite simple. Given such simplicity, I believe the model could easily produce different outputs under perturbed conditions. To strengthen your claim, it would be important to evaluate whether GOHST can induce other forms of hallucination beyond object hallucination using data that requires more complex captioning, rather than relying on binary (Yes/No) responses. In addition, while the inclusion of CHAIR evaluation is a positive step, it does not yet demonstrate that the method helps mitigate other categories of hallucination.
> > >
> > > Furthermore, I personally remain unconvinced about the practical utility of stress tests constructed from artificially perturbed images. In real usage scenarios, it seems highly unlikely that typical MLLM users would input such perturbed images. Likewise, it is not immediately clear what a realistic threat model would look like for an adversary intentionally crafting perturbations to induce object hallucination.
> > >
> > > Given these considerations, I will slightly increase my score to a 4.

---

> > > > ### Author Response · Authors · 2025-12-02
> > > >
> > > > We thank the reviewer for raising their score and for the thoughtful follow-up. We address the two remaining concerns below.
> > > >
> > > > **1. Scope beyond object hallucination.**
> > > > We agree that the current paper primarily focuses on object-existence hallucination. The additional attribute and relation experiments included during the rebuttal phase were intended to illustrate GHOST’s extensibility, not to claim comprehensive coverage of hallucination forms. We select object grounding as the central focus because it is a foundational component of multimodal reasoning, and failures at the object level often propagate into more complex hallucinations. As noted in the paper, GHOST is diagnostic, and the corrective setup we present is a small proof-of-concept demonstrating the potential of GHOST-generated data to reduce hallucinations, rather than a full corrective pipeline.
> > > >
> > > > **2. Practical utility of the stress tests.**
> > > > Our aim is not to model typical end-user inputs or define an adversarial threat model. Instead, GHOST is designed as a diagnostic tool for model developers: the synthetic images surface systematic shortcut behaviors that natural images often fail to reveal. These controlled perturbations expose semantic dependencies underlying hallucination, and our experiments show that these shortcuts can be mitigated through simple fine-tuning. Such insights can inform future model design, robustness evaluation, and training methodology. Additionally, similar to DASH, GHOST samples can be used to search over large natural-image collections to identify nearest-neighbor cases that exhibit similar visual patterns.

---

### Official Review · Reviewer_ZKuB · 2025-11-03

**Soundness:** 3
**Presentation:** 3
**Contribution:** 3
**Rating:** 6
**Confidence:** 4

**Summary:**

The paper introduces GHOST. It first optimizes a CLIP image embedding. It keeps the image close to the source. It pushes away from the target object. It also raises the model’s chance of saying “Yes” to the object query. A small mapper links CLIP space to the vision tokens of the MLLM. After optimization, the method uses unCLIP to render the final image. The images do not contain the object, yet many MLLMs claim the object is present. The method beats a prior approach by a large margin. The authors also use the images to fine-tune models and reduce hallucinations.

**Strengths:**

- The pipeline is simple, modular and well motivated.

- The success rates are strong across many MLLMs. The transfer rate to closed models is also high.

- Preliminary mitigation results via fine-tuning suggest utility beyond red-teaming.

**Weaknesses:**

- The pipline heavly relianced on a single detector (OWLv2); possible false-negatives not quantified; no ensemble verification.

- Limited baseline coverage beyond DASH; no ablations versus strong pixel-space attacks adapted to MLLMs. It should include more public hallucination benchmarks.

- The closed-model settings are not fully described. The compute cost and runtime are not reported. Please ddd default hyperparameters in the main text.

- Minors writing issues (Also in Appendix)

**Questions:**

What detector threshold did you use?

Did you try an ensemble of detectors?

What are the default hyperparameters for the main results?

What are the prompts and decoding settings for closed models?

For reasoning models, when does a “Yes” in thinking flip to “No” in the final answer?

---

> ### Author Response · Authors · 2025-11-21
> **Response to Reviewer ZKuB**
>
> We thank the reviewer for finding GHOST simple and well motivated, for noting the high transfer rate to other models, and for recognising the benefit of our fine-tuning results. Below we address the points raised by the reviewer:
>
> ## Reliance on Object Detector
> We would like to clarify that in our method, the object detector is used only as a sanity-check filter to remove images that may still contain the target object. The core optimization losses are explicitly designed to eliminate object-specific information from the CLIP embedding itself. Therefore, the pipeline does not heavily rely on any specific detector. Additionally, to ensure that our results are not driven by false negatives from the detector, we performed a human evaluation confirming that the generated samples indeed do not contain the target object; details are provided in Section 5.1.
>
> ## Ensemble of Detectors
> Following the reviewer's suggestion, we conducted an additional experiment using an ensemble of four detectors, covering both open-vocabulary and closed-set families: OWLv2, GroundingDINO, YOLO-World, and YOLOv8. A generated sample was filtered out if any of the detectors assigned confidence ≥ 0.5 that the object was present.
>
> We evaluated 300 samples across 5 object classes on Qwen model as victim, and report the results below:
>
> | Object        | ASR  | Total Images | Successful Samples |
> |---------------|------|--------------|--------------------|
> | Boat          | 0.16 | 279          | 45                 |
> | Bottle        | 0.25 | 271          | 68                 |
> | Clock         | 0.15 | 290          | 44                 |
> | Suitcase      | 0.29 | 280          | 81                 |
> | Traffic light | 0.28 | 299          | 84                 |
>
> ### Overall Ensemble-Filtered Performance
>
> | Metric                   | Value |
> |--------------------------|-------|
> | Total successful samples | 322   |
> | Total images             | 1419  |
> | Global ASR               | 22.7% |
>
> These results confirm that GHOST is not tied to a particular detector: even under much stricter settings, the ASR remains comparable to our original performance. It is emphasized that we are filtering out the samples here if 'any' detector assigns confidence of ≥ 0.5 to the object's presence.. Together with GHOST’s efficiency, this demonstrates that the method can be applied flexibly (whether with different detectors or none) and run at scale to surface hallucination-inducing examples.
>
> We will include this ensemble-detector experiment and the above discussion in the revised paper.
>
> ## Object Detector Threshold
> For the main experiments, we used a detector threshold of 0.5, consistent with the hyperparameters reported in Appendix F.3.
>
> ## Baseline Coverage
> Our work targets an underexplored setting (inducing semantic, object-level hallucinations by injecting spurious visual cues), so extensive baseline coverage is inherently limited. Among existing methods, DASH is the closest and most recent state-of-the-art approach operating in a similar problem space.
>
> Prior pixel-level MLLM attacks are fundamentally different in both objective and mechanism. These methods focus on preserving the original image under strict LPIPS or imperceptibility constraints, producing subtle perturbations designed to be human-invisible. In contrast, GHOST introduces semantic, interpretable perturbations by adding object-free but human-perceptible visual cues that intentionally shift scene semantics to diagnose over-reliance on spurious features.
>
> ## Other hallucination benchmarks
> To address the reviewer's concern, we re-ran and updated our fine-tuning results by adding the CHAIR hallucination benchmark, following the setup and prompts from Li et al. [1]. GHOST fine-tuning yields consistent improvements on CHAIR as well. We will include all evaluation and implementation details in the revised version.
>
> |        | POPE | CHAIRs | CHAIRi | captions | vqa2  |
> |--------|------|--------|--------|----------|-------|
> | BaseQwen | 88.7 | 3.8    | 3      | **89.5**     | **72.8** |
> | +ft      | **93.2** | **2.9**    | **2.6**    | 89.4     | 71.5 |
>
> [1] Li et al., Evaluating Object Hallucination in Large Vision-Language Models.

---

> ### Author Response · Authors · 2025-11-21
> **Response to Reviewer ZKuB**
>
> ## Runtime Ablations
> Following the reviewer’s comment, we provide additional runtime measurements for GHOST. We evaluated six object categories with approximately 400 samples per category on a single A100 GPU using Qwen as the victim model. The detailed per-class statistics are shown below:
>
> | Category      | Total Runtime (s) | Total Runtime (min) | Samples | Avg / Sample (s) | Min (s) | Max (s) |
> |---------------|-------------------|----------------------|---------|-------------------|---------|---------|
> | Boat          | 3372.20           | 56.20               | 378     | 8.59              | 2.23    | 20.04   |
> | Bottle        | 3664.77           | 61.08               | 364     | 9.73              | 2.24    | 19.92   |
> | Carrot        | 3982.47           | 66.37               | 387     | 9.97              | 2.13    | 20.04   |
> | Vase          | 4073.92           | 67.90               | 380     | 10.40             | 2.22    | 20.19   |
> | Suitcase      | 3552.89           | 59.21               | 380     | 9.02              | 2.27    | 20.74   |
> | Traffic Light | 2979.33           | 49.66               | 398     | 7.18              | 2.24    | 19.60   |
>
> Across all categories, GHOST requires 7–10 seconds per sample, including both embedding optimization and final generation process. This efficiency comes from operating directly in the embedding space, which yields a short optimization trajectory. For reference, DASH reports approximately 50 seconds per image for PaLI-Gemma-3B and 60 seconds per image for LLaVA-Next-7B, even when using a distilled diffusion model. Thus, GHOST is roughly 5–7× faster per sample. We will include these in the revised paper as well.
>
> ## Hyperparameters
> We provide the full configuration used for the main experiments below, consistent with the values reported in Appendix F.3.
>
> ### Default Hyperparameters (Main Results)
>
> | Model | lr  | total steps | $\tau$  | N | Guidance Scale | $\lambda_{clip}$ |$\lambda_{reg}$ | OD thr. | t  | num inf. |
> |-------|-----|--------------|-----|---|----------------|------------------|------------------|---------|----|-----------|
> | Qwen  | 0.1 | 100          | 0.8 | 4 | 5.0            | 15.0             | 10.0             | 0.5     | 30 | 50        |
> | LLaVA | 0.1 | 100          | 0.8 | 4 | 5.0            | 15.0             | 10.0             | 0.5     | 30 | 50        |
> | GLM   | 0.2 | 125          | 0.5 | 5 | 5.0            | 0.5              | 1.5              | 0.5     | 30 | 50        |
>
> These hyperparameters were used consistently across all reported experiments unless otherwise specified.
>
> In addition, we include complete hyperparameter configurations for fine-tuning the MLLM (Appendix D.1) and training the MLP (Appendix H.1).
>
> ## Closed-Model Settings
> For GPT-4o and Gemini-2.5-Flash, we query the models through their official APIs, following the OpenAI API documentation for GPT-4o and the Google AI Studio documentation for obtaining a key and interacting with Gemini. Images are passed in their original resolution as base64 (GPT-4o) or raw JPEG bytes (Gemini). Since no decoding parameters are overridden in our code, both APIs run with their default generation settings. The exact prompt used is: “Is there a [object] in the image? Answer yes or no. Explain your answer.”
>
> ## Typos in Appendix
> Thanks for pointing this out, we will fix them in the revised version.
>
> ## Faithfulness of the Reasoning Model
> We did not observe cases in the reasoning model  where the model produced a “Yes” in the internal reasoning trace but a “No” in the final answer. For further clarification: in our experiments with the reasoning model, GHOST optimizes the “Yes” token corresponding to the first token generated after the <think> token. This step directs the internal reasoning process toward predicting the presence of the target object. However, hallucination evaluation is based solely on the final short answer, not on intermediate reasoning tokens.

---

### Author Response · Authors · 2025-12-01

We thank all reviewers for their positive assessment of our work. Reviewers highlighted that GHOST provides a simple, modular, and well-motivated pipeline that generates hallucination-inducing images while preserving the underlying image semantics. They also noted that GHOST reveals shared shortcut behaviors across both open-source and closed-source MLLMs, and that preliminary fine-tuning results show promising reductions in hallucination. Below we summarize the key clarifications, new analyses, and additional experiments added to the revised paper.

1. **Finetuning (Section 6.2 + Appendix J):** We re-ran all fine-tuning experiments and expanded the evaluation to include CHAIR, another hallucination benchmark. The updated results show consistent improvements on both POPE and CHAIR.

2. **Other types of hallucination (Section 6.3 + Appendix H):**  To show GHOST’s potential beyond object hallucination, we added an experiment to test whether GHOST can also induce attribute-level and relation-level hallucinations. Using the same setup as the main experiments, we evaluated five non-object queries and observed an average success rate of 32.2%, comparable to our object-existence settings. Qualitative examples and full details are provided in Section 6.3 and Appendix H.

3. **Runtime (Section 5.1 + Appendix F.8):** We added a runtime analysis showing that GHOST runs in ~10 seconds per sample on a single A100 GPU, about 5× faster than the baseline (DASH) even with its distilled diffusion model.

4. **ObjectNet (Appendix G):** We added experiments on ObjectNet to evaluate GHOST on a different visual dataset. The overall success rate (28%) is comparable to our COCO results (~30%), and qualitative examples are included in Appendix G.

5. **Object Detector (Appendix I):** We clarified how GHOST uses an object detector only as a filtering step and added an ensemble-detector analysis using four detectors (OWLv2, GroundingDINO, YOLO-World, YOLOv8). The results show that GHOST remains effective even under stricter filtering.

6. **Naturalness of GHOST Images (Appendix B):** We added a human evaluation comparing the naturalness of GHOST images with the diffusion baseline generations. Participants rated GHOST images comparably natural to the diffusion baseline.

7. **SSIM (Appendix F.9):** To complement FID and provide an additional perceptual quality metric, we added SSIM (Structural Similarity Index) measurements comparing GHOST, unCLIP, and Stable Diffusion. GHOST achieves higher SSIM, indicating stronger preservation of structural and semantic content.

8. **Higher Resolution (Appendix F.10):** We added experiments at 756×756 resolution and found that runtime increases moderately while success rates remain comparable to the 336×336 baseline.

9. **Limitations (Section 7):** We added the limitations section to better clarify the scope of the current study and align the framing with the provided evidence.

We believe these revisions directly address the main concerns raised across reviews. We also note that **two reviewers had increased their scores during the discussion phase prior to ICLR’s reversion**.

---

### Meta-Review · Area_Chair_DSPn · 2026-01-04

**Summary:**

Contribution Summary:
The paper introduces GHOST, a pipeline for generating hallucination-inducing images by optimizing CLIP embeddings to mislead Multimodal LLMs (MLLMs) while preserving semantic similarity to the original input. It demonstrates high attack success rates, transferability across models (including GPT-4o), and potential as a data source for fine-tuning to mitigate hallucinations.

Reviewer Concerns:
* **Scope and Generalization:** Reviewers noted the method was primarily tested on object-existence hallucinations within COCO classes, questioning its ability to stress-test broader hallucination types (attributes, relations) or diverse domains.
* **Naturalness:** Concerns were raised regarding whether the generated images were sufficiently natural or if they appeared to be out-of-distribution artifacts.
* **Practical Utility:** Reviewer 6FPP questioned the practical threat model, noting it is unclear why users would input such perturbed images in real-world scenarios.
* **Component Reliance:** Reviewers questioned the reliance on specific components, such as the OWLv2 detector (potential false negatives) and the SD-unCLIP diffusion model (limiting applicability to specific architectures).

**Reviewer Concerns:**

Addressed Concerns:
* **Detector Reliance:** The authors addressed the concern about relying solely on OWLv2 by testing an ensemble of four detectors (OWLv2, GroundingDINO, YOLO-World, YOLOv8), showing the attack remains effective (22.7% success rate) even under strict filtering.
* **Runtime and Efficiency:** In response to requests for computational cost analysis, the authors provided detailed runtime metrics (~7-10 seconds per sample), showing GHOST is significantly faster than the DASH baseline.
* **Generalization (ObjectNet):** The authors demonstrated strong attack success rates (0.28) on the ObjectNet dataset to address concerns about being limited to COCO.
* **Image Naturalness:** The authors conducted human evaluations where participants rated GHOST images as comparably natural to the diffusion baseline and provided SSIM metrics to demonstrate structural preservation.
* **Mitigation Evaluation:** The inclusion of the CHAIR benchmark in fine-tuning experiments addressed the request for broader validation of corrective capabilities.

Outstanding Concerns:
* **Limited Scope (Object Hallucination):** Despite adding experiments for attribute/relation hallucinations, Reviewers 6FPP and 8Ved maintained that the method is primarily limited to object-existence hallucinations and felt the broader claims of "stress-testing" were slightly overstated.
* **Practical Utility/Threat Model:** Reviewer 6FPP remained unconvinced about the practical utility of stress-testing with artificially perturbed images in real-world scenarios.
* **Dependency on Embedding-Conditioned Models:** Reviewer 8Ved noted that the reliance on models like SD-unCLIP is a practical limitation that restricts applicability to architectures supporting embedding-level conditioning.

**Reviewer Scores:**

* **Reviewer ZKuB:** Original Score: 6. Likely Final Score: 6.
   The reviewer initially rated it a 6. The authors addressed their specific technical questions regarding detector ensembles, runtime, and baselines comprehensively. Given the authors noted ZKuB's concerns were technical rather than conceptual, a positive score maintain is likely.
* **Reviewer 6FPP:** Original Score: 2. Likely Final Score: 4.
   The reviewer explicitly stated "I will slightly increase my score to a 4" after the rebuttal. They acknowledged the improvements (naturalness/runtime) but retained reservations about the practical utility and scope.
* **Reviewer PFEa:** Original Score: 4. Likely Final Score: 4.
   The reviewer did not respond to the rebuttal, but the authors provided a strong justification for the "human-perceptible" design choice and the architectural necessity of SD-unCLIP. These clarifications addressed the core "soundness" concerns raised in the initial review.
* **Reviewer 8Ved:** Original Score: 4. Likely Final Score: 6.
   The reviewer explicitly stated "I will slightly increase my score" after the authors revised the framing and added experiments.
* **Reviewer GNtm:** Original Score: 6. Likely Final Score: 6.
   The reviewer explicitly stated "I will maintain my score" after the clarifications, noting that most concerns had been addressed.

---

### Decision · Program_Chairs · 2026-01-26

Accept (Poster)